# MOMAGRAPH : STATE-AWARE UNIFIED SCENE GRAPHS WITH VISION–LANGUAGE MODEL FOR EMBODIED TASK PLANNING

**Yuanchen Ju**[*1], **Yongyuan Liang**[*2], **Yen-Jen Wang**[*1], **Nandiraju Gireesh**, **Yuanliang Ju**[3]
**Seungjae Lee**[2], **Qiao Gu**[3], **Elvis Hsieh**[1], **Furong Huang**[†2], **Koushil Sreenath**[†1]
[1]University of California, Berkeley    [2]University of Maryland, College Park
[3]University of Toronto
Project website:  https://HybridRobotics.github.io/MomaGraph/

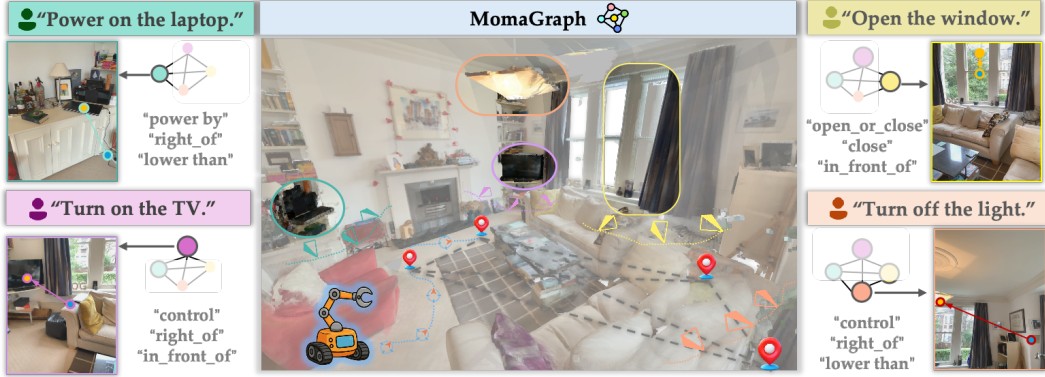

Figure 1: Overview of the **MomaGraph**. Given a task instruction, **MomaGraph** constructs a task-specific scene graph that highlights relevant objects and parts along with their spatial-functional relationships, enabling the robot to perform spatial understanding and task planning.

## ABSTRACT

Mobile manipulators in households must both navigate and manipulate. This requires a compact, semantically rich scene representation that captures *where* objects are, *how* they function, and *which parts* are actionable. Scene graphs are a natural choice, yet prior work often separates spatial and functional relations, treats scenes as static snapshots without object states or temporal updates, and overlooks information most relevant for accomplishing the current task. To address these limitations, we introduce **MomaGraph**, a unified scene representation for embodied agents that integrates spatial-functional relationships and part-level interactive elements. However, advancing such a representation requires both suitable data and rigorous evaluation, which have been largely missing. We thus contribute **MomaGraph–Scenes**, the first large-scale dataset of richly annotated, task-driven scene graphs in household environments, along with **MomaGraph–Bench**, a systematic evaluation suite spanning six reasoning capabilities from high-level planning to fine-grained scene understanding. Built upon this foundation, we further develop **MomaGraph–R1**, a 7B vision–language model trained with reinforcement learning on **MomaGraph–Scenes**. **MomaGraph–R1** predicts task-oriented scene graphs and serves as a zero-shot task planner under a *Graph-then-Plan* framework. Extensive experiments demonstrate that our model achieves state-of-the-art results among open-source models, reaching **71.6%** accuracy on the benchmark (**+11.4%** over the best baseline), while generalizing across public benchmarks and transferring effectively to real-robot experiments.

---

[*] Equal Contribution, [†] Equal Advising

## 1 INTRODUCTION

When mobile manipulators (Qiu et al., 2024; Honerkamp et al., 2024a; Wu et al., 2023; Zhang et al., 2024a) enter household environments, they face the fundamental challenge of understanding how the environment works, which objects are interactive, and how they can be used. In other words, such robots must not only be capable of navigating through the home, but also manipulate objects within it. While navigation requires modeling the overall spatial layout, manipulation demands capturing more fine-grained object affordances (Ju et al., 2024; Zhu et al., 2025). This naturally raises a central question: ***What is the most effective, compact, and semantically rich representation of an indoor scene?*** An intuitive answer is the *scene graph*, which organizes objects and their relationships in a scene through a graph structure (Armeni et al., 2019; Koch et al., 2024a;b) and has shown great potential in various downstream robotic applications (Rana et al., 2023; Werby et al., 2024; Ekpo et al., 2024).

However, existing scene graphs suffer from notable limitations. (1) Their edges typically encode only a single type of relationship, either spatial (Jatavallabhula et al., 2023; Gu et al., 2024a; Loo et al., 2025; Hughes et al., 2024; Rosinol et al., 2021) or functional (Zhang et al., 2025; Dong et al., 2021)(*e.g., a remote controlling a TV, a knob adjusting parameters*). Relying solely on spatial relationships captures geometric layout but overlooks operability, while relying solely on functional relationships ignores spatial constraints, leading to incomplete and less actionable structures. (2) Most existing methods (Wu et al., 2021; Takmaz et al., 2025; Zhang et al., 2021) are limited to static scenes and struggle to adapt to dynamic environments where object positions change or object states change. (3) They lack task relevance, as they fail to emphasize information directly tied to task execution, thereby reducing efficiency and effectiveness. In contrast, cognitive science research (Uithol et al., 2021; Kondyli et al., 2020; Castanheira et al., 2025) shows that human perception in new environments is both dynamic and task-oriented. Humans do not process all information equally; instead, they flexibly adjust their attention according to the current task. This process is similar to browsing a map on an iPad: people first take a broad view to roughly locate the area of interest, and then zoom in to focus on the specific details needed for the task.

Motivated by these insights, we emphasize that ***an ideal scene graph should integrate both spatial and functional relationships, include fine-grained object parts as nodes, making the representation compact, adaptive to dynamic changes, and highly aligned with task instructions, thus providing a more concrete guidance for embodied perception and task planning.***

To achieve this goal, we present **MomaGraph**, a novel scene representation specifically designed for embodied agents. It is the first to unify spatial and functional relationships while introducing part-level interactive nodes, providing a more fine-grained, compact, and task-relevant structured representation than existing approaches. To support this representation, we build **MomaGraph-Scenes**, the first dataset that jointly models spatial and functional relationships with part-level annotations, encompassing multi-view observations, executed actions, and their interactive object parts, and task-aligned scene graph annotations.

Building on this foundation, we propose **MomaGraph-R1**, a 7B vision–language model (VLM) trained with the DAPO (Yu et al., 2025) reinforcement learning algorithm on **MomaGraph-Scenes**. We design a graph-alignment reward function to guide the model toward constructing accurate, task-oriented scene graphs. **MomaGraph-R1** not only predicts scene graphs but also serves as a zero-shot task planner within a *Graph-then-Plan* framework: the model first generates a structured scene graph as an intermediate representation and then performs task planning based on this graph, significantly improving reasoning effectiveness and interpretability.

Despite progress in task-graph planning (Agia et al., 2022), the community still lacks a unified benchmark to systematically evaluate whether and how task-oriented scene graphs improve planning performance. To address this gap, we introduce **MomaGraph-Bench**, a comprehensive evaluation suite that systematically assesses six key reasoning capabilities, spanning from high-level task planning to fine-grained scene understanding.

In summary, our work makes the following key contributions:

- We propose **MomaGraph**, the first scene graph representation that jointly models spatial and functional relationships while incorporating part-level interactive nodes, providing a compact, dynamic, and task-aligned knowledge structure for embodied intelligence.

- We construct **MomaGraph-Scenes**, the first large-scale dataset of richly annotated, task-driven scene graphs in household environments, and build **MomaGraph-Bench**, a unified evaluation suite that systematically measures the impact of scene graph representations on task planning across six core reasoning capabilities.

- We develop **MomaGraph-R1**, a 7B vision-language model that leverages reinforcement learning to optimize spatial–functional reasoning, enabling zero-shot planning in a *Graph-then-Plan* paradigm.

- **MomaGraph-R1** surpasses all open-source baseline models, delivering substantial gains across public benchmarks and translating these improvements into strong generalization and effectiveness in real-world robotic experiments.

## 2 RELATED WORKS

**Scene Graphs for 3D Indoor Scene Understanding.** Scene graphs have emerged as a structured and hierarchical representation in autonomous driving (Zhang et al., 2024b; Greve et al., 2024), robot manipulation (Jiang et al., 2024; Wang et al., 2025; Engelbracht et al., 2024; Jiang et al., 2025; Maggio et al., 2024), and spatial intelligence (Yin et al., 2025; Zemskova & Yudin; Liang et al., 2025a;b) community. They function not only as a means of scene representation but also as a critical bridge between spatial understanding (Cao et al., 2024; Yang et al., 2024; Gu et al., 2024b) and action planning. We focus on the household scenes. However, existing works often focus on a single type of scene graphs. For example, ConceptGraphs (Gu et al., 2024a) primarily model spatial layouts, representing object instances and their geometric relations in an open-vocabulary manner. While spatial graphs (Honerkamp et al., 2024b; Yan et al., 2025) provide useful geometric and semantic grounding, they overlook how objects can functionally interact with one another. Conversely, functional graphs (Li et al., 2021; Dong et al., 2021; Zhang et al., 2025) highlight object affordances and control relations but do not capture the overall spatial structure. Relying solely on either spatial or functional graphs leads to incomplete and less actionable representations. This motivates us to build **MomaGraph**, which unifies spatial and functional relationships, incorporates part-level nodes, and explicitly models state changes, providing a more comprehensive foundation for task planning.

**Zero-shot Embodied Task Planning with VLMs.** VLMs (OpenAI, 2023; Team et al., 2025; Ahn et al., 2022) have gained significant attention in robotic task planning (Niu et al., 2024; Yue et al., 2024; Lu et al., 2023; Liang et al., 2024; Guo et al., 2024) due to their powerful capabilities in processing multimodal inputs, such as images and language instructions. However, when directly used as task planners, VLMs (Huang et al., 2023; 2024; Ahn et al., 2022; Zheng et al., 2025a; Yang et al., 2025) often suffer from sensitivity to visual noise and shallow semantic grounding; more fundamentally, their lack of structured object–relationship representations necessitates extracting or constructing more effective representations from the same visual inputs to support accurate and reliable high-level planning. Prior approaches such as SayPlan (Ahn et al., 2022) assume access to a reliable 3D scene graph, which is often unrealistic in practice. To overcome this gap, we propose the *Graph-then-Plan* strategy, which first generates task-specific scene graphs as an intermediate structured representation before high-level planning. By explicitly modeling objects and their relations, this approach significantly improves the accuracy and robustness of task planning. Unlike prior graph-then-plan methods (Dai et al., 2024; Ekpo et al., 2024) that either assume reliable scene graphs or treat graph construction and planning as separate modules, our approach enables a single VLM to jointly generate structured, task-oriented scene graphs and perform high-level planning.

## 3 PRELIMINARY FINDINGS AND MOTIVATION EXPERIMENTS

To ground our analysis, before the full evaluations we perform two motivating experiments on the **MomaGraph-Bench**. These comparisons are designed to validate our motivation and design principles, and to reveal why our proposed model is essential for embodied task planning. In this section, we aim to answer the following questions.

### 3.1 ARE VLMS RELIABLE FOR DIRECT PLANNING WITHOUT SCENE GRAPHS?

To examine whether direct planning from visual inputs is reliable even for strong closed-source VLMs, we design controlled evaluations on real-world household tasks such as *"Open the window"*

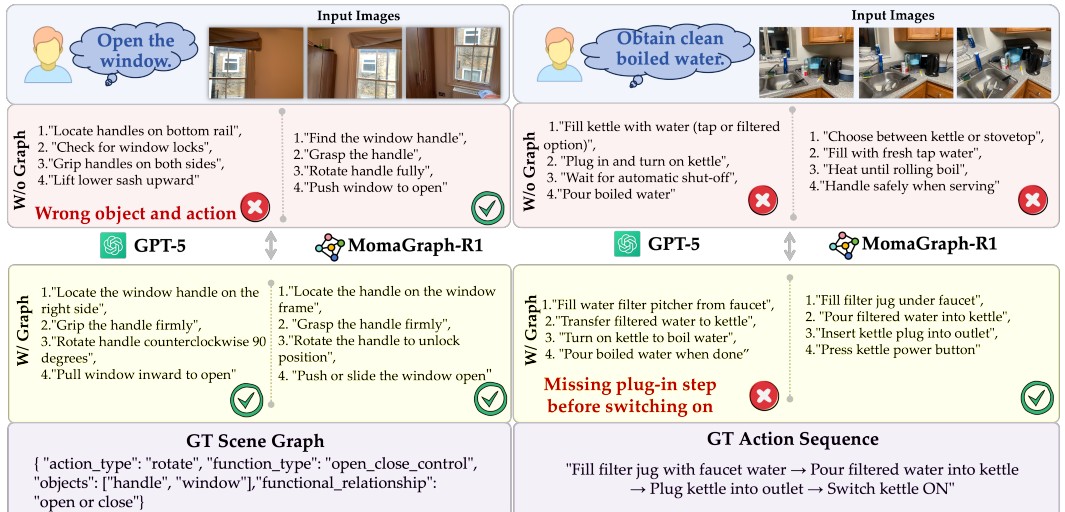

Figure 2: Direct planning often fails even for strong closed-source models like GPT-5, producing wrong actions or missing key steps, while our **Graph-then-Plan** approach with structured scene graphs enables accurate and complete task sequences aligned with ground truth.

and *"Obtain clean boiled water"*. In these scenarios, models must reason over functional relationships, spatial constraints, and multi-step dependencies (e.g., plug-in before activation, filtration before boiling). As shown in Fig. 2, despite their scale, closed-source VLMs like GPT-5 produce incorrect or incomplete plans, missing prerequisite steps, or misidentifying interaction types. In contrast, our **Graph-then-Plan** approach, which first generates a task-specific scene graph and then performs planning, consistently produces correct and complete action sequences aligned with ground-truth logic. This demonstrates that incorporating structured scene representations significantly enhances planning accuracy and robustness beyond what direct planning can achieve.

> **Preliminary Findings 1**
>
> - *In contrast to directly relying on vision-language models for task planning from raw scene images, our **Graph-then-Plan** strategy—which incorporates task-oriented scene graph generation as an intermediate structured representation prior to high-level planning, substantially improves both the accuracy and robustness of task planning.*

## 3.2 ARE SINGLE-RELATIONSHIP GRAPHS ADEQUATE FOR EMBODIED AGENTS?

To ensure a fair comparison, we retrain our model using the same graph structure as in `MomaGraph`, but constrain the edge types to encode only a single kind of relation—either spatial or functional. This setup allows us to isolate the effect of relation types while keeping the graph topology consistent, thereby directly examining whether single-relation representations are sufficient for task planning. To ensure this finding generalizes beyond one specific architecture, we evaluate this comparison across different base models using the same dataset and experimental configurations. As demonstrated in Table 1, both `MomaGraph-R1` (trained from Qwen-2.5-VL-7B) and LLaVA-Onevision consistently show superior performance with unified spatial-functional scene graphs compared to single-relationship variants, supporting our hypothesis that integrated representations are essential for effective embodied task planning. Detailed training methodology is described in the Sec. 4.2.

Table 1: Comparison between `MomaGraph-R1` and LLaVA variants across task tiers.

| Models | T1 | T2 | T3 | T4 | Overall | Models | T1 | T2 | T3 | T4 | Overall |
|---|---|---|---|---|---|---|---|---|---|---|---|
| MomaGraph-R1 (Spatial-only) | 69.1 | 67.0 | 58.4 | 45.4 | 59.9 | LLaVA-Onevision (Spatial-only) | 63.4 | 56.7 | 59.7 | 36.3 | 54.0 |
| MomaGraph-R1 (Functional-only) | 71.4 | 65.8 | 63.6 | 59.0 | 64.9 | LLaVA-Onevision (Functional-only) | 65.1 | 61.7 | 55.8 | 45.4 | 57.0 |
| MomaGraph-R1 (Unified) | **76.4** | **71.9** | **70.1** | **68.1** | **71.6** | LLaVA-Onevision (Unified) | **68.6** | **62.9** | **67.5** | **56.5** | **66.0** |

> **Preliminary Findings 2**
>
> - *Graph representations that rely solely on spatial relationships or solely on functional relationships are insufficient. For embodied agents, **a unified representation that jointly models both spatial and functional relationships** provides a more complete and effective foundation for perception and action.*

## 4 METHOD

### 4.1 MOMAGRAPH DEFINITION

Given a single indoor room, the agent receives as input a set of *multi-view images* $\{\mathcal{I}_i\}_{i=1}^n$ and a natural language instruction $\mathcal{T}$. The objective is to construct an *instruction-conditioned*, **task-oriented scene graph** $\mathcal{G}_\mathcal{T} = (\mathcal{N}_\mathcal{T}, \mathcal{E}_s^\mathcal{T}, \mathcal{E}_f^\mathcal{T})$. Here, $\mathcal{N}_\mathcal{T}$ denotes the set of nodes representing objects relevant to task $\mathcal{T}$. $\mathcal{E}_s^\mathcal{T}$ encodes the *spatial relationships* among these nodes, and $\mathcal{E}_f^\mathcal{T}$ captures their *functional relationships*. This task-oriented scene graph provides a minimal yet sufficient structured representation that grounds the instruction $\mathcal{T}$ in the observed scene and facilitates downstream embodied task planning. Both $\mathcal{E}_s^\mathcal{T}$ and $\mathcal{E}_f^\mathcal{T}$ are modeled as directed edges, pointing from the *triggering object* to the *affected object*.

### 4.2 VLMS LEARN SCENE GRAPH REPRESENTATIONS WITH REINFORCEMENT LEARNING

Existing open-source VLMs have demonstrated limited capability in generating accurate task-oriented scene graphs $\mathcal{G}_\mathcal{T}$ from multi-view observations $\{\mathcal{I}_i\}_{i=1}^n$ and natural language instructions $\mathcal{T}$. VLMs do not form structured spatial-functional representations or reason effectively about task-relevant object relationships needed for embodied tasks. To go further, we want to know: ***Can reinforcement learning teach VLMs to build more precise and task-relevant scene graph representations with `MomaGraph`?***

Reinforcement learning offers a more principled approach by encouraging the model to explore, reason, and iteratively refine its representations through outcome-driven feedback. Rather than replicating memorized patterns, RL enables models to discover effective strategies for constructing task-relevant scene graphs through structured thinking and reasoning. We apply the DAPO (Yu et al., 2025). The key innovation lies in our carefully designed **graph-based reward function** $\mathcal{R}(\mathcal{G}_\mathcal{T}^{\text{pred}}, \mathcal{G}_\mathcal{T}^{\text{gt}})$, where $\mathcal{G}_\mathcal{T}^{\text{pred}}$ and $\mathcal{G}_\mathcal{T}^{\text{gt}}$ denote the predicted and ground truth task-oriented scene graphs, respectively, which evaluates how well predicted graphs embody these principles through three key components.

**Action type prediction.** Given the task instruction $\mathcal{T}$, we ensure correct prediction of the required action type through $R_{\text{action}} = \mathbb{I}[a_{\text{pred}} = a_{\text{gt}}]$, where $a_{\text{pred}}$ and $a_{\text{gt}}$ denote the predicted and ground truth action types, respectively.

**Spatial-functional integration on edges.** We jointly evaluate both spatial relationships $\mathcal{E}_s^\mathcal{T}$ and functional relationships $\mathcal{E}_f^\mathcal{T}$ within each edge, where $\mathcal{E}_{\text{pred}}^\mathcal{T}$ and $\mathcal{E}_{\text{gt}}^\mathcal{T}$ represent the predicted and ground truth edge sets:

$$R_{\text{edges}} = \frac{1}{|\mathcal{E}_{\text{gt}}^\mathcal{T}|} \sum_{e_j \in \mathcal{E}_{\text{gt}}^\mathcal{T}} \max_{e_i \in \mathcal{E}_{\text{pred}}^\mathcal{T}} S_{\text{edge}}(e_i, e_j) \tag{1}$$

where $S_{\text{edge}}(e_i, e_j)$ measures semantic similarity between edges $e_i$ and $e_j$ based on their spatial and functional relationship labels.

**Node completeness.** We compute intersection-over-union similarity for task-relevant objects in $\mathcal{N}_\mathcal{T}$, where $\mathcal{N}_\mathcal{T}^{\text{pred}}$ and $\mathcal{N}_\mathcal{T}^{\text{gt}}$ denote the predicted and ground truth sets of task-relevant nodes: $R_{\text{nodes}} = \frac{|\mathcal{N}_\mathcal{T}^{\text{pred}} \cap \mathcal{N}_\mathcal{T}^{\text{gt}}|}{|\mathcal{N}_\mathcal{T}^{\text{pred}} \cup \mathcal{N}_\mathcal{T}^{\text{gt}}|}$.

The final reward function integrates these task-oriented design principles with format validation and length control, where $R_{\text{format}}$ ensures valid JSON structure and $R_{\text{length}}$ penalizes overly verbose outputs:

$$\mathcal{R}(\mathcal{G}_\mathcal{T}^{\text{pred}}, \mathcal{G}_\mathcal{T}^{\text{gt}}) = w_a \cdot (R_{\text{action}} + R_{\text{edges}} + R_{\text{nodes}}) + w_f \cdot R_{\text{format}} + w_l \cdot R_{\text{length}} \tag{2}$$

where $w_a$, $w_f$, and $w_l$ are hyperparameters controlling the relative importance of each component.

This reward design directly implements our core insight: scene graphs must simultaneously capture spatial layout ($\mathcal{E}_s^{\mathcal{T}}$) and functional relationships ($\mathcal{E}_f^{\mathcal{T}}$) while remaining tightly coupled to task requirements ($\mathcal{T}$). With RL training on **MomaGraph-Scenes**, we develop **MomaGraph-R1**, a 7B vision-language model built on Qwen2.5-VL-7B-Instruct (Qwen, 2025), which learns to generate compact, task-relevant representations that provide concrete guidance for embodied planning.

We demonstrate that RL significantly enhances both the effectiveness and generalizability of open-source VLMs for scene graph generation in the following section. This aligns with broader findings that combining structured scene representations with reasoning consistently improves VLM scene understanding. Critically, **MomaGraph-R1** achieves robust performance across diverse environments and task configurations, enabling practical deployment in unseen embodied scenarios.

### 4.3 STATE-AWARE DYNAMIC SCENE GRAPH UPDATE

In realistic environments, multiple objects of the same category may coexist, and their task-related correspondences are often initially *uncertain*. Take Figure 3 as an example, a kitchen stove may have several knobs, but only one controls the burner required for the current cooking task. Simply relying on visual appearance is insufficient to determine the correct functional relationship. In this work, we do not focus on the agent's interaction policy; instead, our emphasis lies on *how to capture and incorporate observed state changes in the environment* into the scene graph to resolve such ambiguities.

Formally, at time step $t$, the task-oriented scene graph is represented as:

$$\mathcal{G}_{\mathcal{T}}^{(t)} = \left(\mathcal{N}_{\mathcal{T}}^{(t)}, \mathcal{E}_s^{\mathcal{T},(t)}, \mathcal{E}_f^{\mathcal{T},(t)}\right), \qquad (3)$$

where $\mathcal{N}_{\mathcal{T}}^{(t)}$ denotes the set of task-relevant candidate objects, $\mathcal{E}_s^{\mathcal{T},(t)}$ encodes their spatial layout, and $\mathcal{E}_f^{\mathcal{T},(t)}$ captures *hypothesized* functional relationships, which may initially include one-to-many mappings.

After the agent executes an action $a_t$ and observes the new environment state $s_{t+1}$, the scene graph is refined as:

$$\mathcal{G}_{\mathcal{T}}^{(t+1)} = \mathcal{U}\left(\mathcal{G}_{\mathcal{T}}^{(t)}, a_t, s_{t+1}\right), \qquad (4)$$

Figure 3: **MomaGraph** captures state changes in the environment and dynamically updates the task-specific scene graph accordingly, enabling the graph to evolve as interactions occur and reflecting updated spatial–functional relationships.

where the update function $\mathcal{U}(\cdot)$ removes inconsistent hypotheses and strengthens confirmed correspondences based on the observed state transition. As illustrated in Fig. 3, if rotating a specific knob ignites the burner while others have no effect, the functional edge [control] between that knob and the burner is established, while edges from other knobs are pruned. This process enables the scene graph to evolve from ambiguous, one-to-many hypotheses into a compact, *state-aware dynamic representation* with unique and reliable object-to-object correspondences.

## 5 DATASET AND BENCHMARK

### 5.1 MOMAGRAPH-SCENES DATASET

Existing scene graph datasets for 3D indoor environments are often constrained to a single relationship: some focus exclusively on *spatial layouts* of objects (Armeni et al., 2019; Koch et al., 2024b), while others emphasize *functional interactions* (Dong et al., 2021; Zhang et al., 2025). However, these scene graph representations that are restricted to a single relationship type are insufficient for embodied agents, as task execution in household environments requires reasoning about both *where objects are* and *how they can be used*. To address these limitations, we introduce **MomaGraph-Scenes**, the first dataset designed to provide a more comprehensive and task-relevant scene representation. **MomaGraph-Scenes** jointly encodes *spatial relationships* and *functional relationships*, covering **9 spatial relationship types and 6 functional relationship**

**types**, explicitly representing interactive elements such as handles and buttons. Our dataset consists of approximately 1,050 task-oriented subgraphs and 6278 multi-view RGB images, collected from a combination of manually collected real-world data, re-annotated existing datasets (Zhang et al., 2025; Delitzas et al., 2024), and simulated environments built with AI2-THOR (Kolve et al., 2017). These samples span more than **350 diverse household scenes** and encompass **93 distinct task instructions**. Compared with prior datasets, our annotations are significantly more detailed, and capturing interaction semantics at both the object and part levels. This broad coverage ensures rich variability in scene layouts, object configurations, and interaction types, supporting robust learning and evaluation of embodied reasoning.

### 5.1.1 DATASET ANNOTATION

**Multi-View Observation.** The multi-view images provided for each graph are not constrained to always contain every relevant object within each single view. We also do not impose restrictions on the number of viewpoints or their exact configurations. This flexible setup better reflects realistic perception conditions, where embodied agents must reason across partial and diverse observations to build consistent scene graph representations.

**Task Instruction.** It is worth noting that the task instructions in our dataset do not explicitly mention all the objects required to accomplish the task. Instead, they are expressed in simple and natural forms (e.g., "Fill the bathtub"), where the relevant objects such as the *bathtub*, *faucet*, and *button* must be inferred by the model. This design encourages the model to learn how to ground natural instructions into the appropriate set of objects and relationships, rather than relying on object names being explicitly stated.

**Nodes.** $\mathcal{N}_\mathcal{T}$ primarily consists of the objects necessary to accomplish the instruction. When the task execution requires interacting with specific parts, the graph may additionally include *part-level interactive elements* (e.g., handles, knobs, or buttons). For example, for the instruction "Open the fridge," $\mathcal{N}_\mathcal{T}$ includes both the *fridge* and its *handle*; for the instruction "Turn on the light," $\mathcal{N}_\mathcal{T}$ consists of the *switch* and the *ceiling light*.

**Edges.** Edges in the task-oriented scene graph capture both *functional* and *spatial* relationships between nodes.

- **Functional Relationships.** We define a functional relationship as the ability of one object to change the state of another object. In indoor environments, common tasks can be broadly categorized as *Parameter Adjustment, Device Control, Open/Close the Cabinet or Door, Water Flow Control, Power Supply, and Assembly*. Accordingly, we identify six major types:

  [OPEN OR CLOSE], [ADJUST], [CONTROL], [ACTIVATE], [POWER BY], [PAIR WITH].

  Notably, [PAIR WITH] does not alter the internal state of objects but instead modifies their spatial configuration, which is essential for assembly tasks (Qi et al., 2025). Since such tasks are critical for robotic interaction and task planning, we explicitly include [PAIR WITH] as a functional relationship. Through this definition, our dataset extends beyond physical and electronic interactions to encompass fine-grained reasoning about assembly and pairing, enhancing its utility for downstream action execution and planning.

- **Spatial Relationships.** Capture geometric dependencies between objects and parts. The dataset primarily annotates:

  **Directional:** [LEFT OF], [RIGHT OF], [IN FRONT OF], [BEHIND], [HIGHER THAN], [LOWER THAN].

  **Distance-based:** [CLOSE], [FAR], [TOUCHING].

  These annotations provide the rich context necessary for reasoning about layout, reachability, and interaction feasibility.

## 5.2 MOMAGRAPH BENCHMARK AND EVALUATION

We introduce `MomaGraph-Bench`, the first benchmark that jointly evaluates fine-grained scene understanding and task planning abilities across diverse levels of difficulty. Our design principle for `MomaGraph-Bench` is to evaluate whether advances in scene understanding provide tangible improvements in downstream task planning and reasoning. Our evaluation framework examines six essential reasoning capabilities in four tiers of difficulty levels: (1) *Action Sequence Reasoning*, (2)

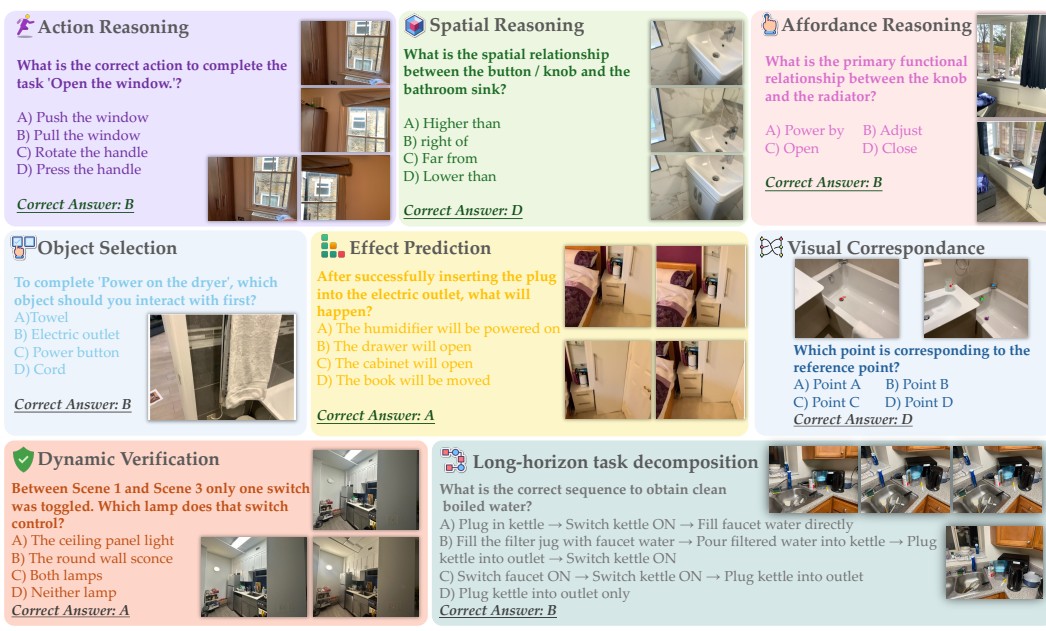

Figure 4: Examples of evaluation Multi-Choices VQA tasks in the `MomaGraph-Bench`. We showcase example questions covering six core reasoning capabilities. Beyond these core capabilities, we further design tasks on *Dynamic Verification* and *Long-horizon Task Decomposition* to evaluate temporal reasoning and multi-steps planning.

*Spatial Reasoning*, (3) *Object Affordance Reasoning*, (4) *Precondition & Effect Reasoning*, (5) *Goal Decomposition*, and (6) *Visual Correspondence* (with concrete examples shown in Fig. 4).

`MomaGraph-Bench` is formulated as a multi-choice VQA task which comprises 294 diverse indoor scenes with 1,446 multi-view images, featuring 352 task-oriented scene graphs spanning 1,315 instances that range from simple step object manipulation(Tier 1) to complex multi-step replanning (Tier 4) scenarios (detailed breakdown in Appendix A.4). `MomaGraph-Bench` offers the most comprehensive assessment for embodied agents' capacity to generalize across tasks and scenarios. To ensure that the evaluation truly reflects generalization rather than memorization, all scenarios are drawn from entirely unseen environments.

# 6 EXPERIMENTS

## 6.1 BENCHMARK EVALUATION FOR EMBODIED TASK PLANNING

We compare the performance of our `MomaGraph-R1` with other models across all task tiers in `MomaGraph-Bench` to rigorously assess embodied planning, including state-of-the-art closed source models (Claude-4.5-Sonnet, GPT-5, Gemini-2.5-Pro) and leading open source models (InstructBLIP, LLaVA-V1.5, DeepSeek-VL2, InternVL2.5, LLaVA-OneVision, Qwen2.5). We further examine whether *Graph-then-Plan* brings performance gains by evaluating each model under two controlled settings: (i) *Direct Plan (w/o Graph)*: the model is directly evaluated on task planning in `MomaGraph-Bench` using multi-view observations and instructions; (ii) *Graph-then-Plan (w/ Graph)*: the model first generates a task-oriented scene graph $\mathcal{G}_\mathcal{T}$, capturing nodes, spatial and functional edges, and action types, and then performs task planning conditioned on the graph.

### 6.1.1 RESULT ANALYSIS.

The results in Table 2 yield several key insights:

**(1) Effectiveness of Graph-then-Plan.** Across all models, the *w/ Graph* setting consistently outperforms the *w/o Graph* baseline, demonstrating that explicitly structuring task-oriented scene graphs provides a tangible benefit for downstream planning. This validates our central hypothesis that disentangling scene representation from action generation improves reasoning reliability.

Table 2: Performance comparison on the **MomaGraph-Bench**. We report accuracy (%) across four tiers (T1–T4) and the overall score, with and without graph-based reasoning.

| Type | Models | Params | MomaGraph Benchmark | | | | | | | | | |
| | | | Tier 1 | | Tier 2 | | Tier 3 | | Tier 4 | | Overall | |
| | | | w/o Graph | w/ Graph | w/o Graph | w/ Graph | w/o Graph | w/ Graph | w/o Graph | w/ Graph | w/o Graph | w/ Graph |
| Closed Source | ✳ Claude-4.5-Sonnet | - | **77.3** | **83.7** | **67.0** | **70.3** | 69.7 | 72.3 | **65.2** | **69.5** | **69.8** | **73.9** |
| | 🌀 GPT-5 | - | 77.3 | 79.8 | 63.4 | 68.2 | **70.8** | **75.0** | 54.5 | 63.6 | 66.5 | 71.6 |
| | ◆ Gemini-2.5-Pro | - | 76.6 | 79.0 | 65.8 | 69.5 | 67.5 | 72.7 | 60.8 | 65.2 | 67.6 | 71.6 |
| Open Source | 🐬 InstructBLIP-7B | 7B | 43.1 | 44.1 | 42.6 | 41.4 | 38.6 | 36.3 | 31.8 | 36.3 | 39.0 | 39.5 |
| | 🔥 LLaVA-V1.5-7B | 7B | 51.0 | 53.4 | 46.3 | 48.7 | 40.2 | 36.3 | 38.9 | 40.9 | 44.1 | 44.8 |
| | 🐋 DeepSeek-VL2 | 4.5B | 54.2 | 56.9 | 51.2 | 53.6 | 61.8 | 61.3 | 40.9 | 45.4 | 52.0 | 54.3 |
| | 🐌 InternVL2.5-8B | 8B | 53.6 | 51.0 | 51.2 | 53.0 | 55.8 | 59.7 | 33.3 | 40.9 | 48.4 | 51.1 |
| | 🌋 LLaVA-Onevision-7B | 7B | 60.0 | 63.8 | 52.4 | 56.0 | 58.4 | 59.2 | 43.4 | 43.4 | 53.5 | 55.6 |
| | 🦅 Qwen2.5-VL-7B-Instruct | 7B | 62.1 | 66.3 | 58.5 | 58.5 | 51.9 | 57.1 | 56.5 | 59.0 | 57.2 | 60.2 |
| | 🐉 **MomaGraph-R1(Ours)** | 7B | **70.2** | **76.4** | **65.8** | **71.9** | **63.6** | **70.1** | **60.8** | **68.1** | **65.1** | **71.6** |

Table 3: Performance comparison on the BLINK and **MomaGraph-Bench**. By enforcing multi-view consistency, our method significantly improves correspondence reasoning across all open-source models.

| Model | 🌀 GPT-5 | | 🌋 LLaVA-Onevision | | 🦅 Qwen2.5-VL-7B-Instruct | | 🐋 DeepSeek-VL2 | | 🐉 **MomaGraph-R1** | |
| | BLINK | MomaGraph-Bench | BLINK | MomaGraph-Bench | BLINK | MomaGraph-Bench | BLINK | MomaGraph-Bench | BLINK | MomaGraph-Bench |
| Results | 66.1 | 81.2 | 59.7 | 70.7 | 58.7 | 72.7 | 57.4 | 68.4 | 63.5 | 77.5 |

**(2) Competitiveness of MomaGraph-R1.** Our **MomaGraph-R1** achieves performance on par with closed-source giants like Claude-4.5-Sonnet and GPT-5, while clearly surpassing all leading open-source VLMs. Notably, **MomaGraph-R1** delivers a $+11.4\%$ relative improvement over its base model (Qwen2.5-VL-7B) under *w/ Graph*, highlighting the effectiveness of reinforcement learning with graph-based rewards.

**(3) Scalability with Task Complexity.** As task complexity increases from Tier 1 to Tier 4, the performance of most open-source baselines drops sharply, reflecting their limited ability to generalize to multi-step reasoning. In contrast, **MomaGraph-R1** exhibits a much smaller degradation, preserving strong performance in Tier 3 and Tier 4. This indicates superior scalability to long-horizon planning scenarios, a crucial capability for embodied agents.

**(4) General Trend Across Communities.** Closed-source models still maintain the highest absolute performance, benefiting from larger-scale pretraining and proprietary data. However, the consistent gap reduction achieved by **MomaGraph-R1** shows that reinforcement learning with graph-structured intermediate representations can substantially narrow the divide, offering a practical path toward competitive open-source systems.

### 6.2 Benchmark Evaluation for Visual Correspondence

As the model learns scene representations from multi-view observations, it exhibits an emergent ability of *cross-view consistency* , which can reason about the same point across different viewpoints. This capability is most evident in visual correspondence tasks. As shown in Table 3, we compare model performance on visual correspondence tasks from public benchmark BLINK Fu et al. (2024) and our **MomaGraph-Bench**. Scene graph representations enhance performance universally by reducing VLM hallucinations in visual perception. By prompting models to first generate structured scene graphs (*w/ Graph*) and then answer questions in single-turn interactions, we force them to explicitly reason about spatial and functional relationships between objects before answering. We primarily evaluate perception on multi-view reasoning and visual correspondence tasks from BLINK, as well as multi-view correspondence in **MomaGraph-Bench**. Our **MomaGraph-R1** achieves state-of-the-art performance among open-source VLMs, leading by 3.8% on BLINK and 4.8% on our correspondence benchmark compared to the best competing open-source models. These results confirm that **MomaGraph-R1** enables more nuanced and detailed perception of complex indoor scenes, effectively mitigating hallucinations and enabling more reliable scene perception.

### 6.3 Real Robot Demonstrations

**Setup.** To validate the effectiveness of our model in real-world settings, we deploy on the RobotEra Q5, a bimanual humanoid platform with a mobile base. An Intel RealSense D455 camera is mounted

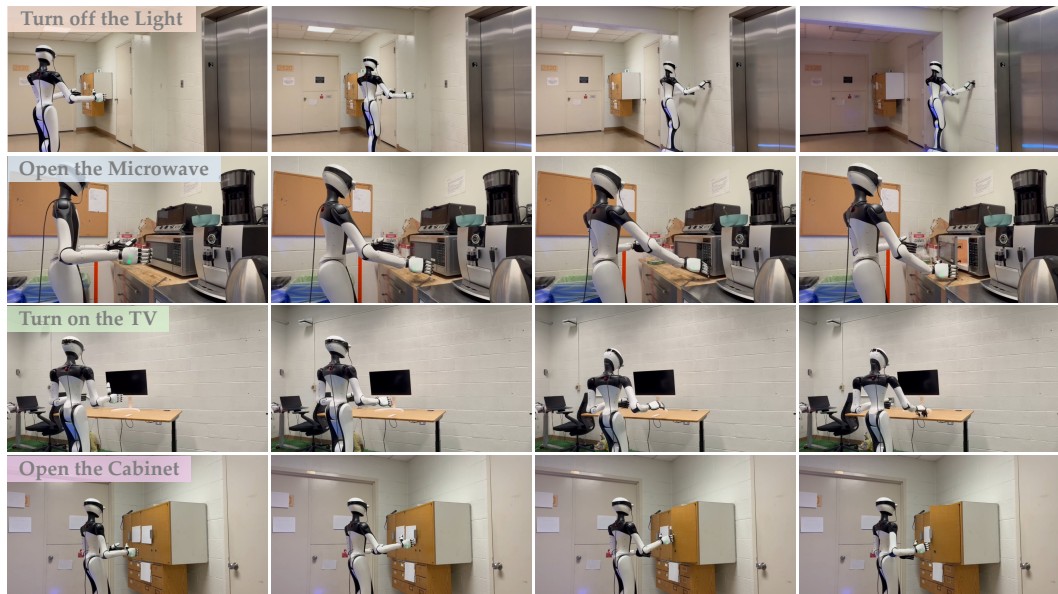

Figure 5: Real Robot experiments on the RobotEra Q5 with a D455, demonstrating four household tasks that require spatial, functional, and part-level interactive elements reasoning for task execution.

to enhance RGB-D perception. Importantly, all evaluation scenes are *unseen*, ensuring that performance reflects true generalization.

**Tasks.** We design four representative tasks (Figure 5), consisting of two *local* interactions (e.g., opening a cabinet, opening a microwave) and two *remote* interactions (e.g., turning on the TV, turning off a light).

**Deployment.** Prior to execution, the robot performs active perception by adjusting its head pose to acquire multi-view observations. `MomaGraph-R1` processes these observations together with the task instruction to generate a task-specific subgraph, which explicitly encodes the relevant objects and their spatial–functional relationships, see more deployment details in B.3. Following the *Graph-then-Plan* paradigm, `MomaGraph-R1` then functions as a task planner, producing a structured action sequence. These specifications are subsequently instantiated as low-level trajectories through a library of parameterized primitive skills. We note that the primitive skills are task-specific and derived from teleoperation data for each scenario; the primary contribution of this work lies in the high-level planning and scene graph generating enabled by `MomaGraph-R1`.

**Summary.** Our real-world evaluations show that `MomaGraph-R1` delivers robust scene understanding and task planning even in unseen scenarios, while remaining directly compatible with standard mobile humanoid systems. This combination underscores the strength of our model and its practicality for real-world deployment.

## 7 CONCLUSION

This work addresses to the fundamental limitations of existing scene graphs for embodied agents: reliance on a single type of relationship, inability to adapt to dynamic environments, and lack of task relevance. To overcome these issues, we introduce `MomaGraph`, a novel scene representation that unifies spatial and functional scene graphs with interactive elements. To learn this representation, we construct a large-scale dataset `MomaGraph-Scenes` and propose `MomaGraph-R1`, a 7B VLM trained with reinforcement learning, which predicts task-oriented scene graphs and serves as a zero-shot task planner under a *Graph-then-Plan* framework. Furthermore, we design the `MomaGraph-Bench`, a comprehensive benchmark that rigorously evaluates both fine-grained reasoning and high-level planning. Through extensive experiments, we demonstrate that our approach achieves state-of-the-art performance among open source models, remains competitive with closed source systems, and transfers effectively to public benchmarks and real robot experiments. We hope that `MomaGraph` will serve as a foundation for advancing scene representations, fostering stronger connections between the spatial VLM and robotics communities, and ultimately enabling more intelligent and adaptive embodied agents.

## 8 ACKNOWLEDGEMENTS

We would like to express our heartfelt thanks to Chenyangguang Zhang, Prof. Florian Shkurti, and Prof. Tom Silver for their insightful suggestions and constructive feedback. We also thank Guowei Zhang, Yuman Gao, Bike Zhang, Gechen Qu, Lihan Zha, Yuanhang Zhang, and Yu Qi for their valuable assistance in the collection of benchmark data. We thank Robot Era for providing their Q5 Mobile Manipulator for our experiments.

Liang, Lee and Huang are supported by DARPA Transfer from Imprecise and Abstract Models to Autonomous Technologies (TIAMAT) 80321, DARPA HR001124S0029-AIQ-FP-019, DOD-AFOSR-Air Force Office of Scientific Research under award number FA9550-23-1-0048, National Science Foundation NSF-IIS-2147276 FAI, National Science Foundation NAIRR240045, National Science Foundation TRAILS Institute (2229885). Private support was provided by Peraton and Open Philanthropy.

The work by Ju, Wang, and Sreenath was supported by The Robotics and AI Institute.

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

## A   APPENDIX

### A.1   MOMAGRAPH-SCENES DATASET

#### A.1.1   REAL-WORLD DATASET SOURCE AND COLLECTION.

Our dataset is built through a synergistic integration of newly curated data and existing public resources. We manually collected a substantial portion of the data in real-world household environments, capturing diverse interaction scenarios under natural conditions. To further enrich the dataset, we incorporated samples from two public benchmarks, OpenFunGraph (Zhang et al., 2025) and SceneFun3D (Delitzas et al., 2024), both of which contain videos depicting human–object interactions in indoor contexts. From these videos, we carefully curated representative keyframes to derive multi-view RGB observations, ensuring comprehensive coverage of interaction dynamics and spatial variability.

#### A.1.2   SIMULATION DATA COLLECTION

To complement the real-world data, we additionally generated samples within the AI2-THOR simulation environment Kolve et al. (2017). We strategically positioned the embodied agent at diverse, reachable viewpoints and captured multi-view observations from varying perspectives, as illustrated in Fig. 6. Throughout this process, we applied manual post-filtering to exclude non-interactable elements, thereby ensuring that the curated dataset remains focused on actionable objects and emphasizes functional relevance critical for downstream embodied reasoning tasks.

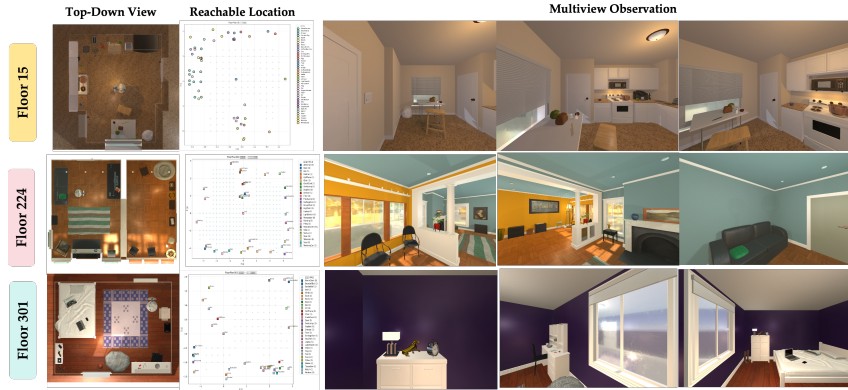

Figure 6: Simulated indoor environments in our benchmark. Each row shows three scenes (*Floor 15*, *Floor 224* and *Floor 301*) with a top-down view of the layout, reachable locations for the robot, and multiview observations from different viewpoints.

#### A.1.3   DATASET ANNOTATION AND FORMAT.

**Annotation and Format.** Each task-oriented subgraph in `MomaGraph-Scenes` is stored in a structured JSON format and linked to its corresponding scene. Annotations include a subgraph identifier, the associated scene identifier, the action type, the functional category, the natural language task instruction, a set of nodes, and a set of edges. Nodes correspond to the *objects or part-level interactive elements* required to accomplish the task, while edges capture both *functional relationships* (e.g., `control`, `open` or `close`) and *spatial relationships* (e.g., `close`, `in_front_of`, `lower_than`).

This example corresponds to the instruction *"Turn on the television"*, where the relevant nodes are the *remote control* and the *TV*, connected by a `control` functional edge and spatial relations `lower_than`, `in_front_of`, and `close`.

In addition, each subgraph is grounded in *multi-view observations*. For every scene, we provide synchronized RGB images captured from multiple viewpoints. This multi-view grounding allows the annotated subgraphs to be consistently aligned with visual evidence, supporting both instruction-conditioned graph prediction from perception and multi-view reasoning tasks.

```
 1  {
 2    "subgraph_id": "da21b9f9-f4fa-4a85-961b-2e2c2e182d3e",
 3    "scene_id": "466828",
 4    "action_type": "press",
 5    "function_type": "device_control",
 6    "task_instruction": "Turn on the television.",
 7    "nodes": [
 8      {"label": "remote control", "id": "f15474de-7b35-4a5e-ac8a-dc02f93960b3"},
 9      {"label": "tv", "id": "91486017-94ce-4788-aabd-0d07262c9bed"}
10    ],
11    "edges": [
12      {
13        "relation_id": "ef3e72fe-ae9f-42e4-9b5a-505b5cb1844a",
14        "functional_relationship": "control",
15        "object1": {"label": "remote control", "id": "f15474de-7b35-4a5e-ac8a-dc02f93960b3"},
16        "object2": {"label": "tv", "id": "91486017-94ce-4788-aabd-0d07262c9bed"},
17        "spatial_relations": ["lower_than", "in_front_of", "close"],
18        "is_touching": false
19      }
20    ]
21  }
```

Figure 7: Example JSON annotation for the task "Turn on the television."

### A.1.4 MULTI-ASPECT STATISTICS OF THE TRAINING DATASET

Our dataset consists of approximately 1,050 subgraphs and 6278 multi-view RGB images, collected across more than 350 diverse household scenes and encompassing 93 distinct task instructions. This broad coverage ensures rich variability in scene layouts, object configurations, and interaction types.

To provide a comprehensive overview of our training data, we present multi-aspect statistics covering scene context, action diversity, functional relationships, and object distributions. As shown in Fig. 8, the dataset spans four common household room types and captures the correspondence between action types and functional categories, reflecting the diversity and richness of real-world manipulation scenarios. Fig. 9 illustrates the distribution of action types across different room contexts, while Fig. 10 summarizes the prevalence of various functional relationships and Fig. 11 summarizes the frequency of object occurrences. Together, these statistics highlight the diversity and task relevance of our dataset, ensuring broad coverage of spatial–functional interactions essential for embodied planning and reasoning.

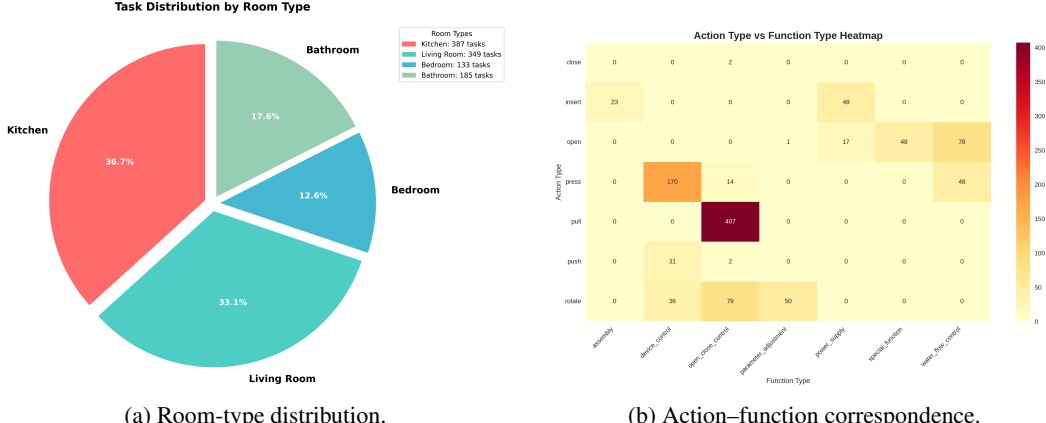

(a) Room-type distribution.       (b) Action–function correspondence.

Figure 8: Dataset statistics: (a) Distribution across four room types; (b)Heatmap showing the correspondence between action types and functional types.

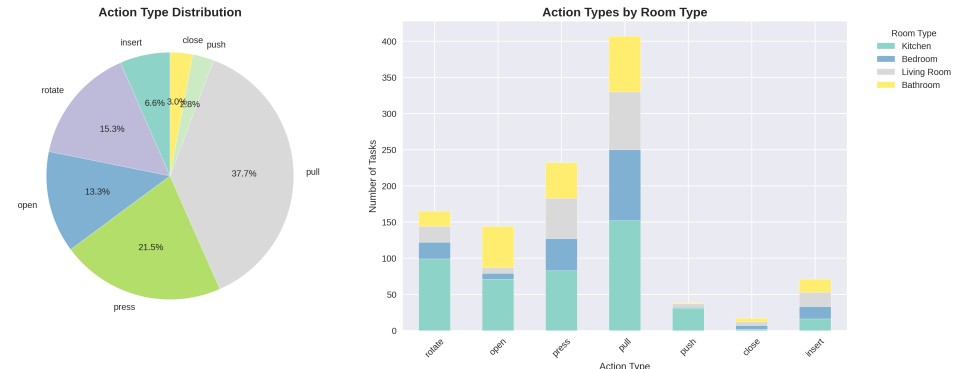

Figure 9: Task distribution across four room types: kitchen, living room, bedroom, and bathroom.

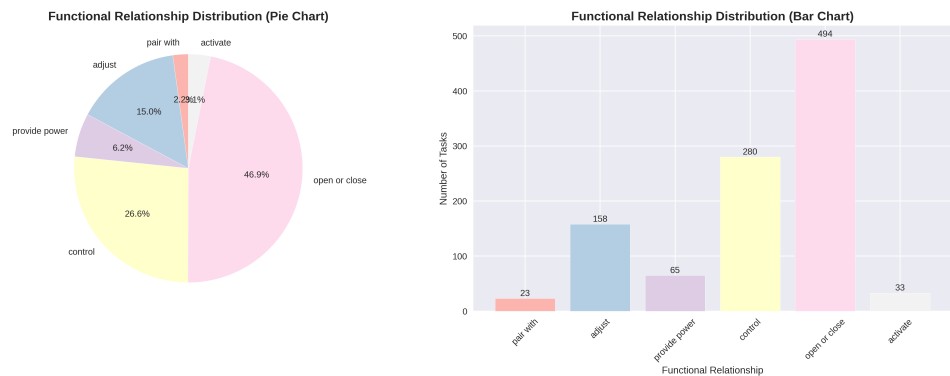

Figure 10: Distribution of functional relationships across all tasks in the dataset.

## A.2  TRAINING DETAILS

We train our model using 8× 80GB A100 GPUs for approximately 13 hours based on the EasyR1 (Zheng et al., 2025b) training framework. The complete training configuration for DAPO algorithm is presented in Table 4.

## A.3  TRAINING CURVE

Figure 12 and  13 shows the training curves during DAPO optimization. The training and validation curves closely align across all metrics, indicating good generalization without significant overfitting. The **overall reward** converges to ∼0.93, while **accuracy reward** stabilizes at ∼0.9. The **format reward** quickly reaches 1.0 within the first 25 steps, showing the model rapidly learns to produce valid JSON-structured outputs.

## A.4  MOMAGRAPH BENCHMARK

### A.4.1  BENCHMARK DESIGN

To rigorously evaluate spatial–functional reasoning and task planning capabilities, we design a comprehensive multi-choice VQA benchmark based on the scenes and tasks in our dataset. Rather than manually crafting all questions, we leverage large vision–language models (VLMs) to generate them in a scalable and diverse manner. Specifically, we provide the model with structured prompts describing the scene images, state-aware scene graph, and task instructions, and instruct it to produce question–answer pairs that probe different reasoning skills, such as spatial relation understanding, affordance inference, precondition reasoning, and goal decomposition. To ensure the reliability and

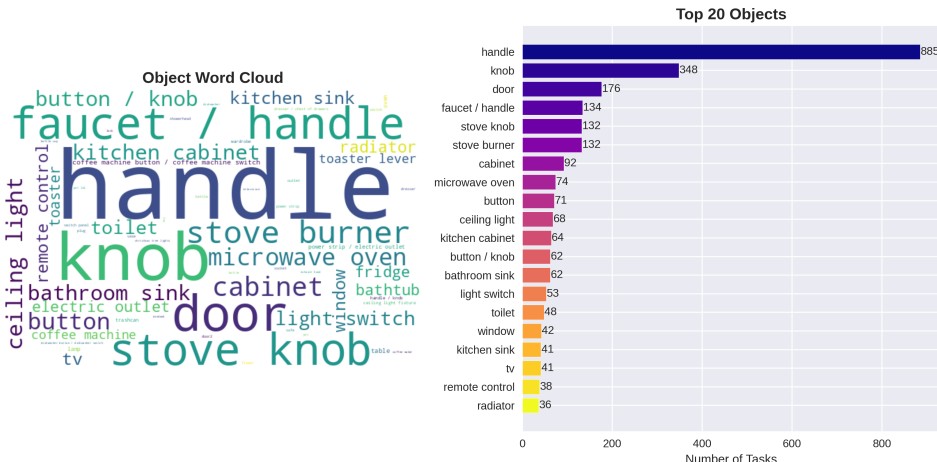

Figure 11: Statistics of object occurrences, highlighting the most frequent objects in tasks.

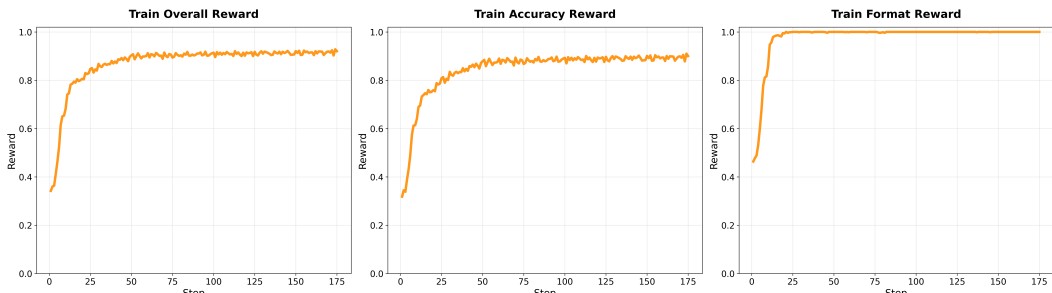

Figure 12: Training reward curves during **MomaGraph-R1** training.

correctness of the benchmark, all generated questions and answers undergo several rounds of manual verification, during which ambiguous or low-quality samples are refined or removed.

Moreover, since the benchmark is formulated as a multi-choice VQA task with clearly defined correct answers, it does not require complex evaluation metrics. Model performance can be directly measured by simple accuracy — i.e., the proportion of correctly answered questions — which provides an intuitive and reliable indicator of spatial–functional reasoning and planning capabilities. This simplicity enables straightforward comparison across models while ensuring that the evaluation remains rigorous and meaningful.

**Data Source and Task Scope.** We leverage long video sequences from SceneFun3D (Delitzas et al., 2024) that capture human-recorded layouts of entire indoor environments, from which key frames are extracted and manually annotated with task-specific graphs. To enhance diversity and coverage, we additionally collect data from real indoor scenes. Our benchmark spans four representative indoor room categories: *bathroom, kitchen, living room, and bedroom*. The task scope is organized into four levels of difficulty:

  **T1** **Single-step actions:** e.g., turning on a light, pulling a drawer, opening a door.
  **T2** **Two complementary steps:** e.g., filling a bathtub by first pressing the drain button and then turning on the faucet.
  **T3** **Multi-step or preconditioned tasks:** e.g., making coffee (pick up a cup → add water → start the coffee machine).
  **T4** **Dynamic verification tasks:** e.g., when the target object is missing, the system must perform *graph-based replanning* and identify *alternative interactive objects*.

- **Action Sequence Reasoning**: examines whether models understand the order and dependency of actions and can plan efficient sequences.

Table 4: DAPO Training Configuration

| Parameter | Value |
|---|---|
| **Model Configuration** | |
| Base Model | Qwen2.5-VL-7B-Instruct |
| Mixed Precision | bfloat16 |
| **Training Setup** | |
| Total Epochs | 25 |
| Training Steps | 175 |
| Actor Global Batch Size | 128 |
| Critic Global Batch Size | 256 |
| Micro Batch Size (Actor) | 1 |
| Micro Batch Size (Critic) | 4 |
| **Optimization** | |
| Learning Rate | 1e-6 |
| Optimizer | AdamW |
| Weight Decay | 0.01 |
| Beta1, Beta2 | 0.9, 0.999 |
| Gradient Clipping | 1.0 |
| **DAPO Algorithm** | |
| KL Coefficient | 0.01 |
| KL Penalty | low_var_kl |
| Disable KL | True |
| Clip Ratio Low | 0.2 |
| Clip Ratio High | 0.28 |
| Clip Ratio Dual | 3.0 |
| **Reward Function** | |
| Format Weight | 0.2 |
| Max Response Length | 2048 |
| Overlong Penalty Factor | 0.5 |
| **Generation Config** | |
| Temperature | 1.0 |
| Top-p | 1.0 |
| Rollout Samples | 5 |

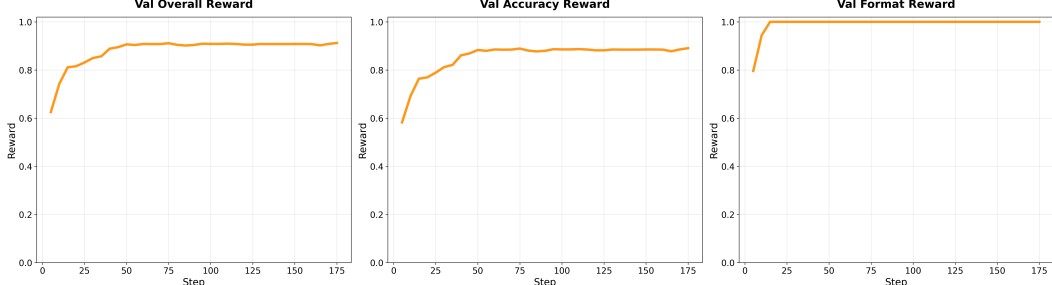

Figure 13: Validation reward curves during `MomaGraph-R1` training.

- **Spatial Reasoning**: focuses on reasoning over spatial relations such as *left_of* or *in_front_of*, judging reachability, and selecting the most suitable object among candidates.

- **Object Affordance Reasoning**: evaluates whether models can infer the functionality of objects (e.g., knobs can be turned, cabinets can be opened), match objects to task requirements, and reason about indirect tool use.

- **Precondition & Effect Reasoning**: assesses whether models understand the preconditions and effects of actions, such as a door needing to be closed before it can be opened, and can predict possible side effects.

- **Goal Decomposition**: measures the ability to break down complex tasks into sub-goals, prioritize them, and determine parallel versus sequential execution strategies.

- **Visual Correspondence (extended capability)**: tests whether models can maintain object consistency across multiple views and integrate information under viewpoint changes.

# B  ADDITIONAL ABLATION STUDIES

## B.1  COMPARISON WITH SFT AND ICL BASELINES

To validate our choice of RL-based training over alternative learning paradigms, we compare our method against two additional baselines:

- **SFT baseline:** We fine-tune Qwen2.5-VL-7B on MomaGraph-Scenes using supervised learning only (without RL), with the same graph-alignment objectives as our full method.

- **ICL baseline:** We evaluate the base model with 3-5 in-context graph examples provided in the prompt (same setting as Qwen2.5-VL-7B-Instruct (w/ Graph) in Table 2 and 3 of the main paper).

| Method | BLINK | MomaGraph-Bench (Overall) |
|---|---|---|
| SFT baseline | 60.4 | 63.9 |
| ICL baseline | 58.7 | 60.2 |
| **RL w/ Graph (Ours)** | **63.5** | **71.6** |

Table 5: Comparison of our RL-based training with SFT and ICL baselines. Our method achieves substantially better performance on both benchmarks.

As shown in Table 5, our RL training method achieves clearly superior performance compared to both the SFT baseline (+3.1 on BLINK, +7.7 on MomaGraph-Bench) and the ICL baseline (+4.8 on BLINK, +11.4 on MomaGraph-Bench). This demonstrates that the RL formulation is crucial for learning high-quality scene graph generation that effectively improves downstream planning performance.

## B.2  REWARD WEIGHT SENSITIVITY STUDY

We follow the original DAPO implementation in the EasyR1 framework for default settings of $w_f$ and $w_l$ in Eq. 2 of the main paper. We conduct a sensitivity study by varying $(w_a, w_f, w_l)$ around the default configuration:

| Setting ID | $w_a$ | $w_f$ | $w_l$ | BLINK | MomaGraph-Bench (Overall) |
|---|---|---|---|---|---|
| A | 0.5 | 0.5 | 0.5 | 61.3 | 68.2 |
| B | 0.7 | 0.3 | 0.5 | 63.1 | 70.9 |
| C | 0.8 | 0.2 | 0.7 | 63.7 | 71.2 |
| **Default** | **0.8** | **0.2** | **0.5** | **63.5** | **71.6** |

Table 6: Sensitivity analysis of reward weights $(w_a, w_f, w_l)$ in our DAPO training. The model's performance remains stable across different weight configurations.

As shown in Table 6, the model's performance remains stable across these weight configurations, with variations of less than 2.4% on BLINK and 3.4% on MomaGraph-Bench. This indicates low sensitivity to reward-weight choices and demonstrates the robustness of our training approach.

### B.3 DETAILED REAL-WORLD DEMONSTRATIONS.

To provide a closer look into the behavior of our system, this section presents fine-grained real-world examples. We illustrate how the model processes raw images captured in realistic household environments, transforms them into task-oriented scene graphs, and generates corresponding planner outputs. These case studies highlight the system's ability to capture subtle details, encode them into structured graphs, and reason over them to produce actionable plans. To validate the effectiveness

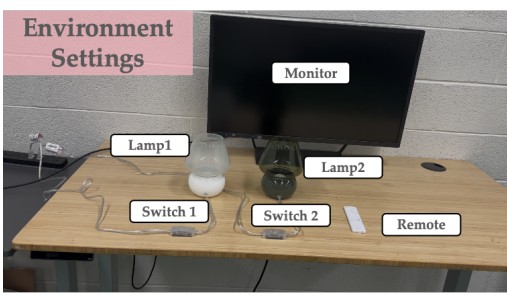 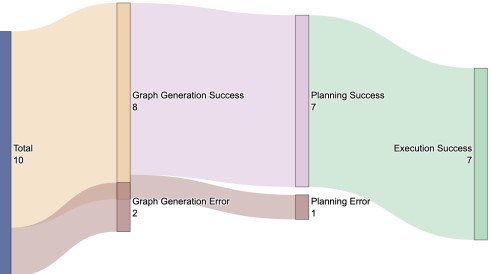

Figure 14: **Quantitative real-robot evaluation.** (a) Environment setup of the real-robot experiment. (b) Failure analysis illustrating success/failure rates across different reasoning stages.

of our approach in real-world settings, we deploy the system on a mobile manipulator to perform a variety of everyday tasks, as shown in Fig. 15. These tasks span multiple functional categories, such as turning off a light, opening a microwave, turning on a TV, and opening a cabinet. In each case, the robot leverages the predicted spatial–functional scene graph to plan and execute a sequence of actions without task-specific fine-tuning. The successful completion of these tasks demonstrates the system's ability to generalize from structured graph representations to real-world interaction scenarios, highlighting its potential for practical household assistance.

### B.4 QUANTITATIVE REAL-ROBOT EVALUATION

To provide rigorous quantitative validation of our system's robustness, we conduct a comprehensive evaluation on a complex multi-step long-horizon task. This evaluation includes success rates and failure analysis across different stages to validate overall system performance under realistic, sequential conditions (see Figure 14).

**Task Setup.** We evaluate the following natural language instruction that requires sequential reasoning and manipulation: *"I need better lighting. Turn on the light closest to the remote so I can find it and turn on the monitor to watch."* To assess system robustness, we conducted 10 experimental trials, changing the camera viewpoint in each trial.

This task requires spatial reasoning (finding the switch and the remote), functional understanding (linking switches, lights, remote, monitor), and state-dependent planning (lighting affects perception). Additionally, there's object uncertainty (multiple similar lamps or switches), complex spatial relations between objects, and sequential manipulation under partial observability.

**Results.** As shown in Figure 14, our system achieves an 80% success rate in graph generation, 87.5% success rate in planning (conditioned on correct graphs), and an overall task success rate of 70% over 10 trials.

The main failure modes were: (1)spatial relation errors or missing nodes during graph generation; and (2) action sequencing errors in the planning phase, suggesting that the system sometimes plans the right actions but in a suboptimal order.

These results demonstrate that `MomaGraph` remains robust across multiple reasoning and execution stages, achieving a 70% overall success rate on a complex multi-step task. This validates the system's reliability under realistic long-horizon conditions where errors can compound across stages.

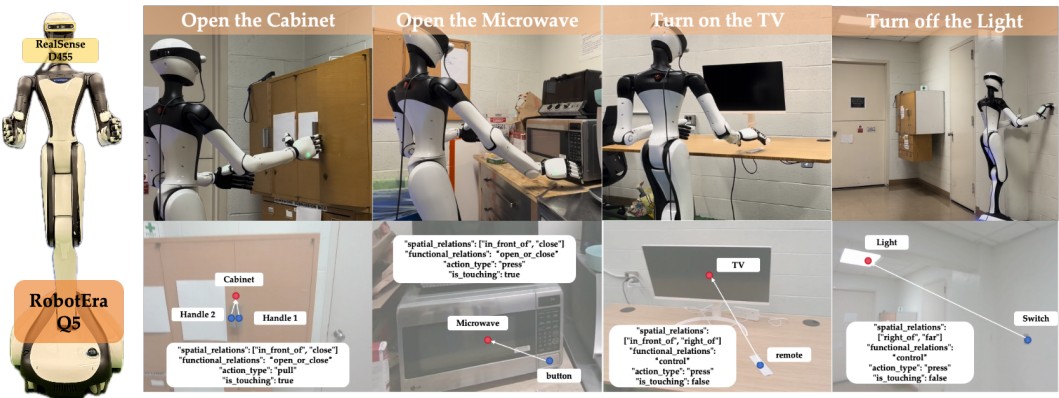

Figure 15: Real-world robot execution of household tasks.

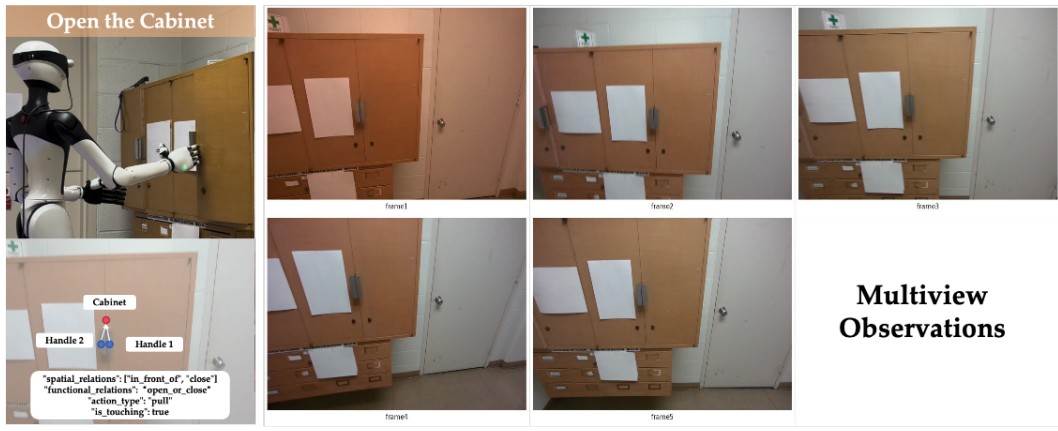

**MomaGraph-R1 Answer:**

Scene Graph Analysis:

First, I understand the task: "Open the cabinet." Then, I identify the key objects needed: two handles and a cabinet. Next, I analyze their spatial relationships: knob1 is in front of and close to the cabinet, while knob2 is in front of and close to the cabinet, requiring a "pull" action for opening/closing operations.

Answer: {"task_instruction": "Open the cabinet.", "nodes": ["handle1", "handle2", "cabinet"], "edges": [{"functional_relationship": "openorclose", "object1": "handle1", "handle2": " cabinet", "spatial_relations": ["in_front_of", "close"], "is_touching": true}, {"functional_relationship": "openorclose", "object1": "handle1", "handle2": " cabinet", "spatial_relations": ["in_front_of", "close"], "is_touching": true}], "action_type": "pull", "function_type": "open_close_control"}

Action Sequence:
1. Approach the target area and identify the cabinet.
2. Locate the handle1, which is in front of and close to the cabinet.
3. Grasp the handle1 and pull it to the right.
4. If the cabinet does not open, locate the handle2, which is in front of and close to the kitchen cabinet.
5. Grasp the handle2 and pull it to the left.
6. Verify that the kitchen cabinet is now open.

Figure 16: Real-world example of **MomaGraph-R1** performing the task "Open the Cabinet." From multiview images, the system generates a scene graph capturing spatial–functional relations and outputs the corresponding action plan.

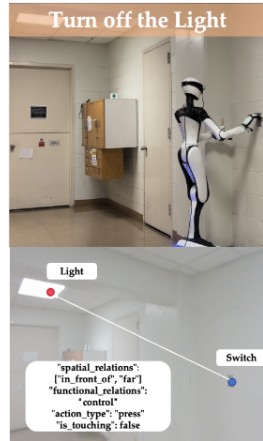 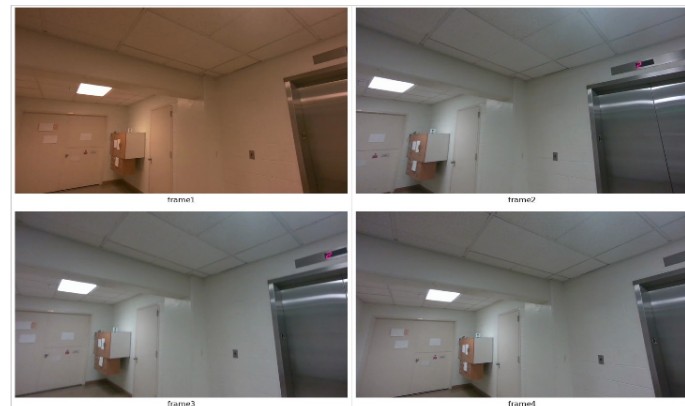

**Multiview Observations**

**MomaGraph-R1 Answer:**

Scene Graph Analysis:

First, I understand the task: "Turn off the light." Then, I identify the key objects needed: a light switch and a light. Next, I analyze their spatial relationships: the light switch is to the right of and far from the light, requiring a "press" action to turn off the light.

Answer: {"task_instruction": "Turn off the light.", "nodes": ["light switch", "light"], "edges": [{"functional_relationship": "control", "object1": "light switch", "object2": "light", "spatial_relations": ["right_of", "far"], "is_touching": false}], "action_type": "press", "function_type": "device_control"}

Action Sequence:

1. Approach the target area: Move towards the light switch, which is located to the right of and far from the light.
2. Identify the key objects mentioned in the scene graph: Locate the light switch and the light.
3. Press the light switch: Use your hand to press the light switch.
4. Verify the task is completed: Check if the light is off by looking at the light.
5. Confirm the task completion: Ensure that the light is off, indicating that the task "Turn off the light" has been successfully completed.

Figure 17: Real-world example of **MomaGraph-R1** performing the task "Turn off the light." From multiview images, the system generates a scene graph capturing spatial–functional relations and outputs the corresponding action plan.

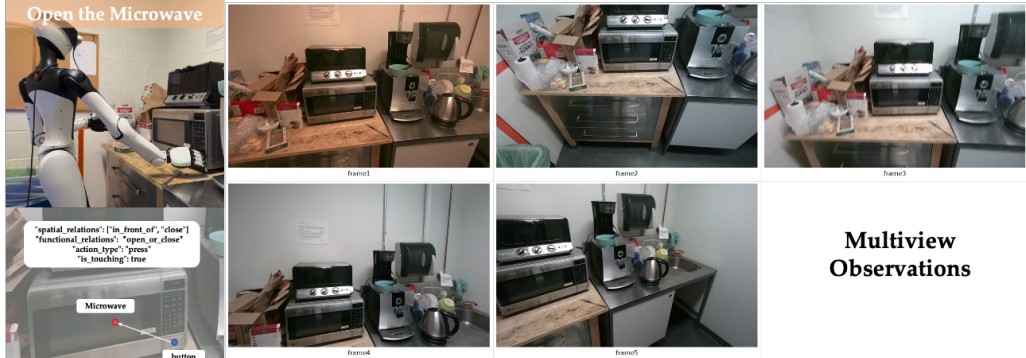

**MomaGraph-R1 Answer:**

Scene Graph Analysis:

First, I understand the task: "Open the microwave." Then, I identify the key objects needed: a microwave and a microwave button. Next, I analyze their spatial relationships: the microwave button is in front of and close to the microwave, requiring a "press" action for opening/closing operations.

Answer: {"task_instruction": "Open the microwave.", "nodes": ["microwave button", "microwave"], "edges": [{"functional_relationship": "openorclose", "object1": "microwave button", "object2": "microwave", "spatial_relations": ["in_front_of", "close"], "is_touching": true}], "action_type": "press", "function_type": "open_close_control"}

Action Sequence:
1. Approach the microwave door, which is in front of the microwave and currently closed.
2. Identify the microwave button and the microwave as the key objects mentioned in the scene graph.
3. Press the button of the microwave with your hand.
4. Verify that the microwave door is fully open.

Figure 18: Real-world example of **MomaGraph-R1** performing the task "Open the microwave." From multiview images, the system generates a scene graph capturing spatial–functional relations and outputs the corresponding action plan.

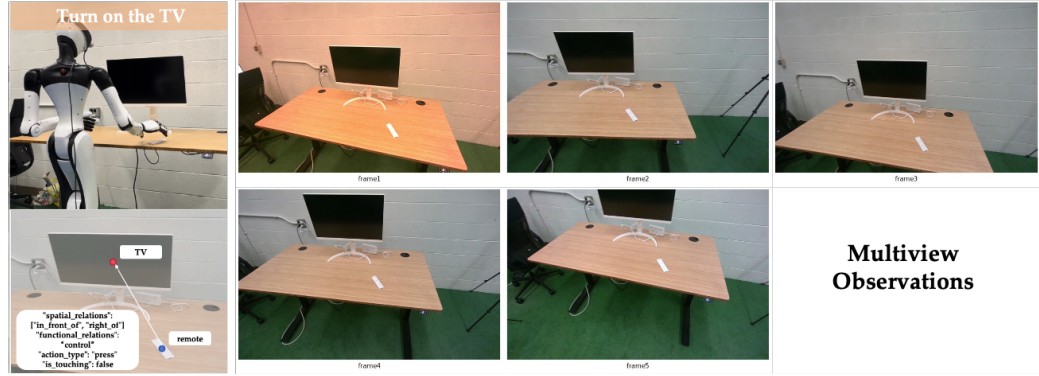

**MomaGraph-R1 Answer:**
Scene Graph Analysis:
First, I understand the task: "Turn on the TV." Then, I identify the key objects needed: a remote and a TV. Next, I analyze their spatial relationships: The remote is to the right of and in front of the TV, requiring a "press" action for activation.

Answer: {"task_instruction": "Turn on the TV.", "nodes": ["remote", "TV"], "edges": [{"functional_relationship": "control", "object1": "remote", "object2": "TV", "spatial_relations": ["in_front_of", "right_of"], "is_touching": false}], "action_type": "press", "function_type": "device_control"}

Action Sequence:
1. Approach the target area: Move towards the TV and the remote, ensuring they are within reach.
2. Identify the key objects mentioned in the scene graph: Locate the remote and the TV.
3. Hold the remote: Use your hand to grasp the remote, ensuring it is in your control.
4. Aim the remote at the TV: Position the remote in front of the TV, making sure it is close enough to the TV's surface for effective control.
5. Press the power button: Locate the power button on the remote and press it firmly. This action should turn on the TV.
6. Verify the task is completed: Check if the TV is turned on by looking for the power indicator light or by attempting to change channels or adjust the volume.

Figure 19: Real-world example of **MomaGraph-R1** performing the task "Turn on the TV." From multiview images, the system generates a scene graph capturing spatial–functional relations and outputs the corresponding action plan.

