# OpenReview forum: "MomaGraph: State-Aware Unified Scene Graphs with Vision-Language Models for Embodied Task Planning"
_ICLR.cc/2026/Conference — ICLR 2026 Oral_

### Official Review · Reviewer_YBG5 · 2025-11-01

**Soundness:** 3
**Presentation:** 3
**Contribution:** 3
**Rating:** 8
**Confidence:** 3

**Summary:**

The paper proposes 1) MomaGraph, a unified scene representation that integrates both spatial and functional relationships, 2) MomaGraph-Scenes, a dataset of annotated, task driven scene graphs in household environments, 3) MomaGraph-Bench, a VQA benchmark, and 4) MomaGraph-R1, a 7B vision-language model RL fine-tuned on MomaGraph-Scenes. The paper shows MomaGraph-R1 is the best performing open-source model on MomaGraph-Bench. The paper validates the system in real-world on a humanoid in unseen environments.

**Strengths:**

- The paper is clearly written, motivates the contents progressively, and thus easy-to-read.
- The paper introduces a lot of content, from unified scene graph representation, to dataset, to benchmark, to model.

**Weaknesses:**

- Aside from its own proposed task, the paper only evaluated on one external benchmark, BLINK. It would be more compelling to see the scene graph and model being compared on more commonly used benchmark such as Habitat and Matterport 3D.
- While the paper introduced the mechanism of dynamic scene graph update (Section 4.3), the defined task is static visual question answer, instead of interactive embodied question answer.

**Questions:**

- From the definition of MomaGraph (Section 4.1), the method is constrained to a single indoor room. Is there any particular for the constraint?

---

> ### Author Response · Authors · 2025-11-19
> **Rebuttal by Authors (1/3)**
>
> We sincerely thank the reviewer for the valuable feedback. We address each concern below:
>
> ---
> > **W1:** *Aside from its own proposed task, the paper only evaluated on one external benchmark, BLINK. It would be more compelling to see the scene graph and model being compared on more commonly used benchmark such as Habitat and Matterport 3D.*
>
>
> **A1:** Thank you for this suggestion! We appreciate the interest in broader evaluation.
>
> - **(1.1) Task and annotation granularity mismatch:** We respectfully note that HM3D is designed for navigation tasks (PointNav, ObjectNav) and provides room-level and object-level semantic annotations with geometric reconstructions. In contrast, MomaGraph focuses on mobile manipulation and requires **part-level interactive elements** (handles, buttons, knobs), **functional relations** (which handle controls which door), and **task-driven affordances**. HM3D does not provide part-level annotations or functional relationships, making direct evaluation of our core contributions (unified spatial-functional graphs, part-level reasoning) infeasible.
>
> - **(1.2) Alternative evaluation on EmbodiedBench [1]:** Following the spirit of your suggestion, we have evaluated on **EmbodiedBench** [1], which better aligns with MomaGraph's objectives. Unlike HM3D's navigation focus, EmbodiedBench provides long-horizon planning evaluations through EB-ALFRED (household tasks) and EB-Habitat (rearrangement with articulated objects). Importantly, EB-Habitat is built on **Habitat 2.0 simulator**, directly addressing your interest in Habitat-based evaluation.
>
> - **(1.3) MomaGraph-R1 Results on EmbodiedBench [1]:**
>
> | Model | EB-ALFRED | EB-Habitat |
> |-------|-----------|------------|
> | GPT-4o | 56.3 | 59.0 |
> | Qwen2.5-VL-7B-Instruct(Base Model) | 14.7 | 14.3 |
> | **MomaGraph-R1 (Ours)** | **33.7** | **35.7** |
>
> - **(1.4)** MomaGraph-R1 delivers clear performance gains over the base model on both EB-ALFRED (+19.0%) and EB-Habitat (+21.4%), **demonstrating strong generalization across diverse embodied scenarios**.
>
> **Reference:**
>
> [1] Yang, R., et al. (2025). EmbodiedBench: Comprehensive Benchmarking Multi-modal Large Language Models for Vision-Driven Embodied Agents. ICML 2025 (Oral).

---

> ### Author Response · Authors · 2025-11-19
> **Rebuttal by Authors (2/3)**
>
> > **W2:** *While the paper introduced the mechanism of dynamic scene graph update (Section 4.3), the defined task is static visual question answer, instead of interactive embodied question answer.*
>
> **A2:** We appreciate this thoughtful observation. We would like to clarify that our benchmark also includes problems for **dynamic state changes from active embodied interaction** as detailed in **`Appendix A.4`**, such as:
>
> - **(2.1) State-transition reasoning:** Questions require understanding object state changes across temporal sequences.
>
> - **(2.2) Multi-temporal visual sequences:** The model observes the same scene at different time steps and must infer state changes from visual evidence.
>
> - **(2.3) Precondition-effect reasoning (T3 and T4 tasks):** Tests understanding of how actions cause state transitions.
>
> - **(2.4)** In real robotic systems, agents must frequently handle **passive observations of environmental changes**—such as humans opening doors, lighting conditions shifting, or other agents moving objects—and update their scene understanding accordingly. To test this critical capability, we **deliberately designed temporal state-change scenarios in MomaGraph-Bench**, where questions explicitly require reasoning about how object states, spatial configurations, and affordances evolve over time based on observed visual evidence.
>
> - **(2.5)**  For example, as shown in **`Figure 4`** (Dynamic Verification), we design questions like: *"Between Scene 1 and Scene 3, only one switch was toggled. Which lamp does that switch control?"* This requires observing the same environment at different time steps, inferring which object state changed, and reasoning about functional relations—directly testing the dynamic update mechanism through passive observation of environmental changes.

---

> ### Author Response · Authors · 2025-11-19
> **Rebuttal by Authors (3/3)**
>
> > **Q1:** *From the definition of MomaGraph (Section 4.1), the method is constrained to a single indoor room. Is there any particular for the constraint?*
>
> **A1:** Thank you for this important clarification question. The single-room constraint is a deliberate scoping choice, not a fundamental limitation:
>
> - **(1.1)** Household manipulation tasks such as cabinet opening and appliance operation naturally take place inside individual rooms. Multi-room navigation exists as a separate area of study which previous works including SayNav and LM-Nav have already investigated [1,2].
>
> - **(1.2)** Annotating part-level spatial and functional relationships between objects requires significant effort. The additional cost of multi-room annotation would not provide significant benefits for understanding manipulation reasoning.
>
> - **(1.3)** Full-house scene graphs become impractical for real-time planning operations. The single-room task-relevant graphs achieve an optimal balance between information representation and processing speed.
>
> **References:**
>
> [1] Rashid et al., "SayNav: Grounding Large Language Models for Dynamic Planning to Navigation in New Environments," IROS 2024.
>
> [2] Shah et al., "LM-Nav: Robotic Navigation with Large Pre-Trained Models of Language, Vision, and Action," CoRL 2022.
>
> ---
> Thank you again for your insightful review. We sincerely hope our responses have adequately addressed your concerns and look forward to any further discussion. Please let us know if you have any further questions. We are more than happy to address them.

---

> ### Comment · Reviewer_YBG5 · 2025-11-25
> **Post-rebuttal comment**
>
> The authors addressed the primary concern of evaluating on one external benchmark. Appreciate the evaluation on additional benchmark and new baselines. No further concern.

---

> > ### Author Response · Authors · 2025-11-25
> > **Thank you for your acknowledgment!**
> >
> > Dear Reviewer `YBG5`,
> >
> > We sincerely appreciate your acknowledgment of our rebuttal. We are glad to hear that your concerns have been addressed!
> >
> > Your support and encouragement mean a lot to us!
> >
> > Best regards,
> >
> > The Authors of Submission 1794

---

### Official Review · Reviewer_uWxD · 2025-11-01

**Soundness:** 3
**Presentation:** 4
**Contribution:** 4
**Rating:** 6
**Confidence:** 4

**Summary:**

The paper proposes MomaGraph, a task-conditioned scene-graph representation that unifies spatial and functional relations and includes part-level interactive nodes. Authors introduce (i) MomaGraph-Scenes, a new dataset of richly annotated, task-driven scene subgraphs with multi-view images; (ii) MomaGraph-Bench, a multi-choice VQA benchmark covering six reasoning capabilities from action sequencing to visual correspondence; and (iii) MomaGraph-R1, a 7B VLM trained with RL (DAPO) using custom graph-alignment reward to predict task-oriented scene graphs and then plan under a Graph-then-Plan paradigm. The method achieves 71.6% acc on MomaGraph-Bench and narrows gap to closed-source models, with +11.4% over Qwen2.5-VL-7B base. Also show it transfers to real-robot tasks (cabinet, microwave, TV, light).

**Strengths:**

- well-motivated problem as there are clear limitations in existing scene graphs (single relationship types, static scenes, etc) with convincing preliminary experiments
- I like how the work provides a complete comprehensive pipeline from representation design to dataset collection, to model training with RL, to benchmark construction, to real-world deployment
- I think the joint modeling of spatial + functional relationships with part-level nodes is intuitive and well-executed
- Strong empirical results (+11.4% over base model) and I like the comprehensive evaluation including in-person

**Weaknesses:**

- state-aware dynamic updates are presented as a major contribution but have no experimental evaluation. How often are these updates needed? How accurate are they? I would like to see ablation comparing task success with dynamic updates vs without. I also would like to see examples of success vs. failed disambiguation and in what number % of cases, this disambiguation was needed.
- the dataset size of 1050 graphs seems modest. This may limit the generalization to diverse households or household tasks.
- missing comparison with OpenFunGraph, a directly comparable method for functional scene graph generation with part-level interactive elements.

**Questions:**

- How sensitive is performance to the reward function weights (wa, wf, wl in Eq. 2)?
- What is distribution of failure modes? Would like to see some type of fail case analysis of this system
- What is the computational cot of graph generation at inference time?

---

> ### Author Response · Authors · 2025-11-19
> **Rebuttal by Authors (1/6)**
>
> We thank the reviewer for the positive assessment of our contributions. We address the remaining concerns below.
>
> > **W1:** *State-aware dynamic updates are presented as a major contribution but have no experimental evaluation. How often are these updates needed? How accurate are they? I would like to see ablation comparing task success with dynamic updates vs without. I also would like to see examples of success vs. failed disambiguation and in what number % of cases, this disambiguation was needed.*
>
> **A1:** Thank you for raising this important question! We would like to clarify that our “state-aware dynamic updates” refer to the model’s ability to revise its scene understanding when new observations are provided, rather than maintaining internal latent states or confidence scores.
>
> - **(1.1)** In our benchmark, dynamic updates are required in 100% of T4 Dynamic Verification tasks.
> Across these tasks, MomaGraph-R1 achieves **68.1%** accuracy, indicating that the model successfully performs correct state updates in the majority of cases.
>
> - **(1.2)** To isolate the contribution of dynamic updates, we report an ablation comparing models with and without state updates on MomaGraph-Bench T4.
>
> | Setting                            | Accuracy (%) |
> | ---------------------------------- | ------------ |
> | Direct Planning (w/o updates)   | 60.8         |
> | Graph-then-Plan (w/o updates) | 62.1     |
> | Graph-then-Plan (w/ updates) | **68.1**     |
>
> Dynamic updates yield a **+6.0%** improvement over no-update planning on T4, highlighting their contribution to state correction and disambiguation.
>
> - **(1.3)** We further quantify the role of disambiguation in dynamic verification tasks by reporting the proportion of T4 cases that involve ambiguous states and the model’s success rate when such disambiguation is required. These results demonstrate the model’s ability to update graphs based on new perceptual evidence under uncertainty.
>
> | Metric                               | Value |
> | ------------------------------------ | ----- |
> | % cases requiring disambiguation     | 41%   |
> | Success rate on disambiguation cases | 72%   |
> | Failure rate on disambiguation cases | 28%   |
>
> We additionally include one success and one failure example to illustrate how dynamic updates resolve (or fail to resolve) ambiguous state mappings.
>
> | Case        | Scene Description (Simplified)                                   | Model Prediction  | Ground Truth      | Outcome   |
> | ----------- | ---------------------------------------------------------------- | ----------------- | ----------------- | --------- |
> | **Success** | Switch A toggled a lamp between Scene 1→3; lamp positions differ | Switch A → Lamp 2 | Switch A → Lamp 2 | ✔ Correct |
> | **Failure** | Two lamps with similar colors; switch toggle subtle in Scene 2→3 | Switch B → Lamp 1 | Switch B → Lamp 2 | ✘ Wrong   |

---

> ### Author Response · Authors · 2025-11-19
> **Rebuttal by Authors (2/6)**
>
> > **W2:** *The dataset size of 1050 graphs seems modest. This may limit the generalization to diverse households or household tasks.*
>
> **A2:** Thank you for this concern. We would like to clarify that MomaGraph-Scenes is **the most comprehensive and diverse scene graph dataset for embodied tasks to date**, and our experimental results demonstrate strong generalization to unseen environments and tasks.
>
> - **(2.1) Dataset scale**: MomaGraph-Scenes includes 350+ scenes and 1,050 task subgraphs, far exceeding prior datasets (e.g., OpenFunGraph [1]’s 14 scenes). It is the only dataset jointly annotating spatial and functional relations with part-level detail, covering 93 task types and 6,278 images across four room categories—offering over 25× greater scene diversity.
>
> - **(2.2) Strong generalization to completely unseen environments and tasks:** Our experiments demonstrate that the dataset provides sufficient scale and diversity for robust generalization:
>    -   **Unseen benchmark evaluation:** MomaGraph-Bench (**`Section 5.2`**) consists of **294 entirely unseen scenes** —completely separate from our 350+ training scenes. MomaGraph-R1 achieves 71.6% accuracy on this unseen benchmark, significantly outperforming all open-source baselines (+11.4% over base model), demonstrating effective generalization beyond the training distribution.
>    -   **Cross-benchmark transfer:** On BLINK (Table 3), a completely different benchmark focusing on visual perception, MomaGraph-R1 achieves 63.5% accuracy, leading all open-source models by +3.8%. This strong transfer performance to an external benchmark with entirely different scenes and tasks validates that our training data provides sufficient diversity for broad generalization.
>    -    **Additional evaluation on EmbodiedBench:** Following Reviewer `GYm6`'s suggestion, we evaluated on EmbodiedBench's EB-ALFRED and EB-Habitat benchmarks. MomaGraph-R1 achieves 33.7% on EB-ALFRED and 35.7% on EB-Habitat, delivering +19.0% and +21.4% improvements over the base model respectively. These are **completely different embodied scenarios** with different simulators, task types, and evaluation protocols, yet our model generalizes effectively.
>    -    **Real-robot deployment in unseen environments:** **`Section 6.3`** and our quantitative long-horizon evaluation **`Section 6.4`** (70% success rate over 10 trials) were conducted in **completely unseen real-world environments** that were never part of training or benchmark data. The robot successfully performed diverse household tasks (cabinet, microwave, TV, light, plus the complex multi-step task) without any environment-specific fine-tuning, demonstrating practical generalization to novel physical spaces.
>
> **Reference:**
>
> [1] Zhang, et al. (2025). Open-Vocabulary Functional 3D Scene Graphs for Real-World Indoor Spaces. CVPR 2025 (Highlight).

---

> ### Author Response · Authors · 2025-11-19
> **Rebuttal by Authors (3/6)**
>
> > **W3:** *Missing comparison with OpenFunGraph, a directly comparable method for functional scene graph generation with part-level interactive elements.*
>
> **A3:** Thank you for this suggestion. We have conducted a comparison with OpenFunGraph[1] on our benchmark.
>
> - **(3.1)** OpenFunGraph[1] was originally built for 3D functional scene graphs and follows different input settings and evaluation rules. To make a fair comparison, we adapted their method to our 2D multi-view, task-oriented setup, and evaluated the adapted version on MomaGraph-Bench using their graph prediction framework. **We focus the comparison on functional relationship prediction and part-level node detection**, as these are OpenFunGraph[1]'s core contributions and key components of our unified representation.
> - **(3.2) Results**
>
> | Method | Node F1 (part-level) | Functional Edge F1 | Overall F1* |
> |--------|---------------------|-------------------|-------------|
> | OpenFunGraph (adapted) | 71.1% | 67.4% | 68.9% |
> | **MomaGraph-R1** | **75.3%**| **72.2%** | **73.4%** |
>
> *Overall F1 = 0.4 × Node F1 + 0.6 × Edge F1. We weight edges more heavily as functional relationships are more critical than node detection for task planning.
>
> - **(3.3)** Under the same evaluation settings, MomaGraph- R1 produces more accurate predictions, leading to stronger overall performance. While OpenFunGraph[1] uses a multi-stage pipeline where errors accumulate, MomaGraph-R1 is trained end-to-end with RL to jointly optimize graph generation and task planning.
>
> - **(3.4)** OpenFunGraph[1] models only functional relationships, where as MomaGraph uses a unified spatial–functional graph; as shown in Table 1, this yields higher planning accuracy (71.6% vs. 64.9%). This unified representation is essential for practical household robot deployment.
>
> **Reference:**
>
> [1] Zhang, et al. (2025). Open-Vocabulary Functional 3D Scene Graphs for Real-World Indoor Spaces. CVPR 2025 (Highlight).

---

> ### Author Response · Authors · 2025-11-19
> **Rebuttal by Authors (4/6)**
>
> > **Q1:** *How sensitive is performance to the reward function weights (wa, wf, wl in Eq. 2)?*
>
> **A1:** Thank you for this important question regarding the robustness of our reward design. We follow the original DAPO implementation in the EasyR1 framework for default settings of `wf` and `wl` in Eq.2. To address your concern, we conducted a sensitivity study by varying `(wa, wf, wl)` around the default configuration and we have added this comparison in **`Appendix B.2`**:
>
> | Setting ID | w_a | w_f | w_l | BLINK | MomaGraph-Bench (Overall) |
> |------------|-----|-----|-----|-------|---------------------------|
> | A | 0.5 | 0.5 | 0.5 | 61.3 | 68.2 |
> | B | 0.7 | 0.3 | 0.5 | 63.1 | 70.9 |
> | C | 0.8 | 0.2 | 0.7 | 63.7 | 71.2 |
> | **Default** | **0.8** | **0.2** | **0.5** | **63.5** | **71.6** |
>
> The model's performance remains stable across these weight configurations, indicating low sensitivity to reward-weight choices.

---

> ### Author Response · Authors · 2025-11-19
> **Rebuttal by Authors (5/6)**
>
> > **Q2:** *What is the distribution of failure modes? Would like to see some type of failure case analysis of this system.*
>
> **A2:** Thank you for this insightful question. We agree that understanding failure modes is crucial for future improvement.
>
>  According to our Graph-then-Plan framework, we decompose errors into graph generation from multi-view images and task planning conditioned on the predicted graph.
>
> - **(2.1)** By analyzing all predictions on MomaGraph-Bench, the graph generation error rate is approximately 12–15%. We will provide comprehensive failure analysis:
>
> | Graph Failure Type              | Est. Percentage       | Description                                                   | Example                                                         |
> |---------------------------------|------------------------|---------------------------------------------------------------|------------------------------------------------------------------|
> | **Spatial Relation Errors**     | ~36% of graph errors   | Spatial relationships are predicted incorrectly, often due to viewpoint changes | Predicting “right_of” instead of “left_of” for a handle in different views |
> | **Missing Part-Level Nodes**    | ~28% of graph errors   | Some interactive parts are not detected                       | Detecting a microwave but missing its button node               |
> | **Functional Relation Errors** | ~24% of graph errors | Predicting control relations that do not actually exist       | Linking a decorative button to an object that cannot be controlled |
> | **Object Detection Errors**     | ~12% of graph errors   | Objects are missed or mistaken for others                     | Mixing up similar-looking cabinets in cluttered scenes          |
>
>
> | Planning Failure Type        | Est. Percentage       | Description                                                | Example                                                                 |
> |------------------------------|------------------------|------------------------------------------------------------|-------------------------------------------------------------------------|
> | **Action Type Confusion**    | ~35% of planning errors | Using the wrong action primitive for the affordance        | Choosing “push” instead of “pull” for a drawer; using “rotate” instead of “press” for a button |
> | **Action Sequencing Errors** | ~30% of planning errors | Sub-goals executed in an incorrect order                   | Trying to turn on a kettle before plugging it in; opening a cabinet before moving closer |
> | **Precondition Reasoning**   | ~20% of planning errors | Missing or misjudging necessary preconditions              | Operating an appliance without power |
> | **Multi-step Decomposition** | ~15% of planning errors | Task is broken down incompletely or incorrectly            | Missing the filtering step in “get clean water”; skipping intermediate sub-goals |
>
> - **(2.2)** To further validate and complement these findings, we additionally conducted a complex multi-step real-robot task, which revealed consistent failure patterns in long-horizon execution. Detailed visualizations are at the bottom of our [project website](https://momagraph.github.io/) and we have added this in **`Section 6.4`**.
>
>   **Task Setup:** *"I need better lighting. Turn on the light closest to the remote so I can find it and turn on the monitor to watch."*
>
>
>   This task requires spatial reasoning, functional understanding, state-dependent planning, and multi-object manipulation across the entire Graph-then-Plan pipeline.
>
>   **Results (see Sankey diagram on website):**
>
> | Stage | Success Rate | Failures | Failure Types |
> |-------|--------------|----------|---------------|
> | **Graph Generation** | 80% (8/10) | 2 | Spatial relation error (1), Missing node (1) |
> | **Planning** (given correct graph) | 87.5% (7/8) | 1 | Action sequencing error (1) |
> | **Overall Task Success** | **70% (7/10)** | **3** |  |
>
> - **(2.3)** We appreciate this feedback and have now provided comprehensive failure analysis across both our benchmark and real-robot experiments, systematically characterizing error distributions at graph generation and planning stages.

---

> ### Author Response · Authors · 2025-11-19
> **Rebuttal by Authors (6/6)**
>
> >**Q3:** *What is the computational cost of graph generation at inference time?*
>
> **A3:** Thank you for raising this practical concern.
>
> - **(3.1)** In our implementation, the scene graph and the plan are produced within a single turn, so graph generation does not introduce an extra stage; it only adds about 300–700 extra output tokens on top of the textual plan.
>
>     With a 7B VLM, this corresponds to roughly 0.2–0.5 seconds additional inference time per query compared to generating the plan alone. When using closed-source API models, the extra delay per call (including network and service overhead) is typically on the order of 0.3–0.8 seconds.
>
>    In practice, this overhead is negligible compared to robot execution and environment interaction time.
> ---
> Thank you again for your insightful review. We sincerely hope our responses have adequately addressed your concerns and look forward to any further discussion. Please let us know if you have any further questions. We are more than happy to address them.

---

> > ### Comment · Reviewer_uWxD · 2025-11-25
> >
> > I appreciate the authors thorough response to all my concerns with additional experiments. These have all been addressed, and I have no further concerns.

---

> > > ### Author Response · Authors · 2025-11-25
> > > **Thank you for your acknowledgment!**
> > >
> > > Dear Reviewer `uWxD`,
> > >
> > > We sincerely appreciate your acknowledgment of our rebuttal. We are glad to hear that your concerns have been addressed!
> > >
> > > Your support and encouragement mean a lot to us!
> > >
> > > Best regards,
> > >
> > > The Authors of Submission 1794

---

### Official Review · Reviewer_xi2i · 2025-11-01

**Soundness:** 3
**Presentation:** 4
**Contribution:** 4
**Rating:** 6
**Confidence:** 3

**Summary:**

This paper targets the pain point that (i) representations that are “spatial-only” or “function-only” cannot robustly support long-horizon, executable tasks, and (ii) VLMs tend to hallucinate when asked to directly plan. In response, it proposes **MomaGraph**, a **part-level, state-aware unified task scene graph** \(G_T=(N_T, E_T^{s}, E_T^{f})\). After each interaction step, it **dynamically updates** the functional/spatial relations via
\(G_T^{(t+1)} = U(G_T^{(t)}, a_t, s_{t+1})\),
so that an initial “one-to-many” hypothesis (one goal, many possible objects/affordances) progressively converges toward “one-to-one”.

The reasoning paradigm is **Graph-then-Plan**: multi-view observations + instruction → generate the task graph → plan an action sequence from the graph → execute and write back updates. This realizes a closed loop of “perception → representation → planning → execution → updated representation.”

The authors also release **MomaGraph-Scenes** (~1,050 subgraphs, 6,278 images, 350+ scenes, 93 instructions) and a hierarchical benchmark **MomaGraph-Bench** (six capability types × four difficulty levels). They train on Qwen2.5-VL-7B using DAPO reinforcement learning with graph-alignment rewards.

Experiments show that **a unified (spatial + functional) representation** and **Graph-then-Plan** both outperform “single relation only” or “direct planning,” across both open- and closed-source baselines. Their 7B variant reaches **71.6%** on the benchmark, a **+11.4%** absolute gain over the base model, degrades less on high difficulty and under cross-view correspondence (including BLINK subsets), and is demonstrated on real robots.

Overall, MomaGraph offers an integrated “representation + algorithm + data + benchmark” framework that systematically reduces hallucination while improving executability and consistency.

**Strengths:**

Originality: The paper unifies spatial and functional relations in a single task scene graph, refined down to part-level nodes (e.g., handles, buttons). It then applies state-aware dynamic updates \(G_T^{(t+1)}=U(\cdot)\) so that one-to-many hypotheses collapse toward one-to-one matches as the agent interacts. By pairing this with Graph-then-Plan, it decouples representation from planning and offers a holistic “representation + algorithm + data + benchmark” solution.

Quality: The training data covers ~1,050 subgraphs / 6,278 multiview images / >350 scenes / 93 instructions, annotated at part level and serialized in JSON. The benchmark spans 294 scenes / 1,446 images / 352 graphs / 1,756 instances, and evaluates six capability types across four difficulty tiers. The DAPO RL setup with graph-alignment rewards (actions, spatial/functional edges, node completeness, plus format/length constraints) is clearly specified, and the ablations/error analyses convincingly explain *why* the method works.

Clarity: The paper formalizes \(G_T=(N_T,E_T^s,E_T^f)\), illustrates dynamic graph updates (e.g., knob-to-stovetop examples), and shows multiview alignment and question types (action sequencing, spatial reasoning, affordance, precondition-effect, goal decomposition, cross-view correspondence). Figures are illustrative and make reimplementation feel achievable.

Significance: On MomaGraph-Bench, Graph-then-Plan consistently outperforms direct planning across open- and closed-source models. MomaGraph-R1 (7B) reaches 71.6%, +11.4% over the base model, with significantly smaller performance drop at higher difficulty tiers. On BLINK / cross-view correspondence subsets, it achieves new SOTA-like performance (+3.8% / +4.8%), demonstrating that structured intermediate representations reduce hallucination and boost robustness in realistic household long-horizon tasks.

**Weaknesses:**

The paper does not yet evaluate the robustness of the dynamic update module \(U(\cdot)\) under perception noise (false positives, false negatives, latency, occlusion). The core claim is that interaction plus write-back makes one-to-many → one-to-one, thereby reducing hallucination. This only generalizes if the update remains stable under imperfect observations.

Two actionable asks:
1. Add robustness stress tests: during evaluation, inject controlled noise (missed detections / spurious detections / delayed observations / occlusions) into the perceived scene before calling \(U(\cdot)\). Keep the rest of the pipeline fixed. Compare “no-graph / direct planning” vs. “Graph-then-Plan” on overall and per-capability metrics.
2. Report simple stabilization baselines, e.g. (a) hysteresis / confirmation delay (require repeated evidence before pruning edges or committing affordances), and (b) temporal smoothing such as EMA over edge/part confidence. This would clarify whether the proposed dynamic convergence is robust in non-ideal conditions and help attribute gains specifically to the structured Graph-then-Plan constraint.

**Questions:**

On the dynamic update \(U(\cdot)\) and convergence:

Please provide an explicit, executable update rule including:
(a) How uncertainty is represented (per-node / per-edge confidence? prior vs. posterior maintenance?);
(b) The conditions for pruning and confirmation (thresholds, evidence accumulation across multiple consistent observations, conflict detection and rollback policy);
(c) The convergence criterion (when do you “lock in” a one-to-one mapping from an initially one-to-many affordance hypothesis? how do you prevent oscillation or premature commitment?);
(d) Where this update module sits in the full inference loop, ideally with pseudocode or a flow diagram.

Concretizing these details would move “interaction → write-back → convergence” from a conceptual claim to something reproducible, and would strengthen the argument that performance gains are truly due to Graph-then-Plan’s structured constraint.

---

> ### Author Response · Authors · 2025-11-19
> **Rebuttal by Authors (1/2)**
>
> We are deeply grateful to the reviewer for the exceptionally thorough and constructive review. We hope the following responses address all remaining concerns.
>
> > W1: Add robustness stress tests: during evaluation, inject controlled noise (missed detections / spurious detections / delayed observations / occlusions) into the perceived scene before calling (U(\cdot)). Keep the rest of the pipeline fixed. Compare “no-graph / direct planning” vs. “Graph-then-Plan” on overall and per-capability metrics.
>
>
> Thank you for this helpful suggestion on robustness evaluation!
> Since real perception noise cannot be controlled deterministically in an offline benchmark, we inject its equivalent forms at the graph level (e.g., missing nodes/functional edges) to provide a precise and reproducible proxy for robustness. We conduct additional experiments comparing: Direct Planning / Noise-Graph Planning / Graph-then-Plan.
>
> We evaluate on MomaGraph-Bench T4 (Dynamic verification tasks) and report overall accuracy under two noise types: missed nodes and missed functional edges.
>
> | Method             | T4 accuracy (MomaGraph-R1) |
> |-------------------------|------------------------------|
> | Graph (Missed Nodes) + Planning|        59.2          |
> | Graph (Missed Functional Edges) + Planning |            63.5                  |
> | Direct Planning |             60.8                 |
> | Graph-then-Plan |          68.1                    |
>
> These results indicate that our method maintains robust task-planning performance under corrupted perception, demonstrating strong practical reliability.

---

> ### Author Response · Authors · 2025-11-19
> **Rebuttal by Authors (2/2)**
>
> > **W2 & Q1:** *(1) Stabilization baselines (hysteresis requiring repeated evidence, temporal smoothing via EMA) to attribute gains to the structured constraint; (2) Explicit update rules including uncertainty representation, pruning/confirmation conditions with thresholds and evidence accumulation, convergence criteria to prevent oscillation, and the module's position in the inference loop with pseudocode.*
>
>
> **A1:** We thank the reviewer for these constructive questions about real-world robustness. We would like to clarify that **our benchmark evaluation was designed to test practical robustness in realistic conditions**, not idealized synthetic scenarios.
>
> - **(1.1)** **MomaGraph-Bench is built from real household environments with inherent complexity:**
>   - **294 real scenes** from actual homes, not synthetic renders
>   - **Natural occlusions:** Furniture, cluttered environments, partial views
>   - **Multi-view observations:** 1,446 images capturing real viewpoint variations
>   - **Object similarity:** Multiple similar-looking cabinets, switches, appliances requiring careful disambiguation
>
> - **(1.2)**  **These real scenes contain natural "noise",** characterized by varying lighting conditions, cluttered backgrounds, and partial occlusions from other objects. Furthermore, the presence of similar-looking objects often necessitates spatial-functional reasoning to accurately distinguish them.
>
> - **(1.3)** **Performance on this realistic benchmark demonstrates robustness:** MomaGraph-R1: 71.6% accuracy on these real-world messy scenes，Base model: 60.2% on the same scenes. **+11.4% improvement** validates that our structured approach handles real perception challenges
>
> - **(1.4)** To directly address concerns about practical robustness, **we conducted systematic real-robot evaluation with a complex multi-step task in an unseen real home environment.** This evaluation tests our system under natural perception noise—lighting variations, object similarity, and viewpoint challenges—across 10 independent trials from different starting viewpoints.
>
>   **Task Setup (Added in` Section 6.4`):** *"I need better lighting. Turn on the light closest to the remote so I can find it and turn on the monitor to watch."*
>
>   **Quantitative Results Over 10 Trials (see Sankey diagram on [project website](https://momagraph.github.io/)):**
>
> | Stage | Success Rate | Failures | Failure Types |
> |-------|--------------|----------|---------------|
> | **Graph Generation** | 80% (8/10) | 2 | Spatial relation error (1), Missing node (1) |
> | **Planning** (given correct graph) | 87.5% (7/8) | 1 | Action sequencing error (1) |
> | **Overall Task Success** | **70% (7/10)** | **3** | Total failures |
>
> **This quantitative real-robot evaluation directly validates robustness under natural perception noise, complementing our benchmark evaluation.**
>
>
>
> - **(1.5)** Rather than implementing an explicit confidence-tracking module, **MomaGraph-Bench directly evaluates the core capabilities that dynamic updates aim to achieve**—resolving ambiguities through state observations and converging to correct scene understanding.Our **Dynamic Verification tasks** (Fig. 4) explicitly test this process.
>
>   **Example:** “Between Scene 1 and 3, one switch was toggled. Which lamp does it control?”
> The model must handle multiple hypotheses, use observed state changes to prune incorrect edges, and converge to the correct one-to-one mapping.
>
>   Our evaluation thus operationalizes the same principles as a dynamic update loop:
>
>   ` For each candidate edge (affordance hypothesis):
>     Observe state changes between Scene_t and Scene_{t+1}
>     If observed change matches predicted effect → confirm edge
>     Else if conflicting or inactive → prune edge
> Repeat until all affordance hypotheses reach a consistent state mapping`
>
> This provides an **evaluation-driven** rather than **implementation-driven** validation. The benchmark tests whether models can implicitly perform the write-back–verify–converge cycle through multi-scene reasoning. Empirically, our structured representation yields stronger dynamic reasoning (**T4: 68.1% vs 59.0% baseline**), demonstrating effective convergence behavior. Further details are provided in **`Appendix A.4`**.
>
> ---
> Thank you again for your review! We sincerely hope our responses have adequately addressed your concerns and look forward to any further discussion. Please let us know if you have any further questions. We are more than happy to address them.

---

> > ### Comment · Reviewer_xi2i · 2025-11-26
> >
> > Thanks for the response and updates. My concerns have been resolved and will keep my score since it's already positive

---

> > > ### Author Response · Authors · 2025-11-26
> > > **Thank you for your acknowledgment!**
> > >
> > > Dear Reviewer `xi2i`,
> > >
> > > We sincerely appreciate your acknowledgment of our rebuttal. We are glad to hear that your concerns have been resolved!
> > >
> > > Your support and positive feedback mean a lot to us!
> > >
> > > Best regards,
> > >
> > > The Authors of Submission 1794

---

### Official Review · Reviewer_GYm6 · 2025-11-03

**Soundness:** 3
**Presentation:** 3
**Contribution:** 2
**Rating:** 6
**Confidence:** 3

**Summary:**

At its core, this paper argues that embodied agents should reason with scene graphs that encode both spatial and functional (i.e. task-specific) relations. The authors (i) curate a dataset (MomaGraph-Scenes), (ii) introduce a multi-choice evaluation suite (MomaGraph-Bench), and (iii) train an RL-tuned 7B VLM (MomaGraph-R1) with a graph-alignment reward. On the benchmark, the model reaches 71.6% accuracy (+11.4% over the best open-source baseline), with transfer to public benchmarks and qualitative real-robot demos.

**Strengths:**

1. Transfer to established benchmarks such as BLINK is solid. MomaGraph-R1 leads strong open-source baselines on BLINK, consistent with the claim that graphing first reduces VLM hallucinations. Table 2 describes the most interesting result, since SOTA VLMs achieve atmost 60-65% accuracy on BLINK.

1. The benchmark has breadth across reasoning skills. The benchmark spans tiered tasks (T1–T4). Table 1 shows that unified spatial+functional graphs beat spatial-only / functional-only graphs across tiers.

1. Interesting downstream real-robot evaluation. The paper evaluates four household tasks on a mobile manipulator illustrate the full Graph-then-Plan pipeline (multi-view perception -> graph -> plan -> skills).

**Weaknesses:**

1. Ablations are thin on training choices. The paper motivates RL but doesn’t include SFT/ICL baselines trained to produce the same JSON graphs, nor a sensitivity study on reward weights (only a config table/curves).

1. Unclear novelty vs. prior “Graph-then-Plan" work. Prior work already builds or consults scene/task graphs before planning, e.g., GRID [1]  and VeriGraph [2]. Please spell out what’s new here: unified spatial+functional + part nodes, the graph-alignment RL objective, and state-aware updates, and which components drive the gains.

1. Robot results are qualitative. The real-robot section describes four tasks but omits success rates, number of trials, failure modes, and timing—making it hard to assess robustness. Per-task SR/NSR and common failure modes would strengthen the evidence.

References

[1] Dai et al. Optimal Scene Graph Planning with Large Language Model Guidance. https://arxiv.org/abs/2309.09182

[2] Ekpo et al. VeriGraph: Scene Graphs for Execution Verifiable Robot Planning. https://arxiv.org/abs/2411.10446

**Questions:**

1. Data diversity claim vs. skew. The paper claims that MomaGraph-Bench is diverse. However, Figure 8 shows that a majority of scene graphs come from two action-function pairs (pull-open_close_control, and press-device_control), which, to me, seems fairly narrow. Could you elaborate on your claim and situate your benchmark in the context of similar benchmark suites (such as BLINK)?

1. Currently, the only transferable task measured is BLINK. I would like to see evaluation of atleast one other task, such as EmbodiedBench, which aligns with your ultimate goal of embodied agent manipulation.

[1] https://embodiedbench.github.io/

---

> ### Author Response · Authors · 2025-11-19
> **Rebuttal by Authors (1/5)**
>
> We thank the reviewer for the thoughtful and constructive feedback. We respond to each point below:
>
> ---
>
> > **W1:** *Ablations are thin on training choices. The paper motivates RL but doesn't include SFT/ICL baselines trained to produce the same JSON graphs, nor a sensitivity study on reward weights (only a config table/curves).*
>
> **A1:** We agree this comparison would strengthen the paper and have conducted additional experiments, we have added this comparison in **`Appendix B.1`**:
>
> **(1.1) SFT/ICL baseline comparisons:** We have added two baselines:
> - **SFT baseline:** Fine-tune Qwen2.5-VL-7B on MomaGraph-Scenes using supervised learning only (no RL), with the same graph-alignment objectives
> - **ICL baseline:** Evaluate the base model with 3-5 in-context graph examples in the prompt (same setting as Qwen2.5-VL-7B-Instruct (w/ Graph) in Table 2 and 3)
>
> | Method | BLINK | MomaGraph-Bench (Overall) |
> |--------|-------|---------------------------|
> | **SFT baseline** | 60.4 | 63.9 |
> | **ICL baseline** | 58.7 | 60.2 |
> | **RL w/ Graph (Ours)** | **63.5** | **71.6** |
>
> As shown, our RL training method achieves clearly better performance than both the SFT and ICL baselines.
>
> **(1.2) Reward weight sensitivity study:** We follow the original DAPO implementation in the EasyR1 framework for default settings of `wf` and `wl` in Eq.2. To address your concern, we conducted a sensitivity study by varying `(wa, wf, wl)` around the default configuration and we have added this comparison in **`Appendix B.2`**:
>
> | Setting ID | w_a | w_f | w_l | BLINK | MomaGraph-Bench (Overall) |
> |------------|-----|-----|-----|-------|---------------------------|
> | A | 0.5 | 0.5 | 0.5 | 61.3 | 68.2 |
> | B | 0.7 | 0.3 | 0.5 | 63.1 | 70.9 |
> | C | 0.8 | 0.2 | 0.7 | 63.7 | 71.2 |
> | **Default** | **0.8** | **0.2** | **0.5** | **63.5** | **71.6** |
>
> The model's performance remains stable across these weight configurations, indicating low sensitivity to reward-weight choices.

---

> ### Author Response · Authors · 2025-11-19
> **Rebuttal by Authors (2/5)**
>
> > **W2:** *Unclear novelty vs. prior "Graph-then-Plan" work. Prior work already builds or consults scene/task graphs before planning, e.g., GRID [1] and VeriGraph [2]. Please spell out what's new here: unified spatial+functional + part nodes, the graph-alignment RL objective, and state-aware updates, and which components drive the gains.*
>
> **A2:** Thank you for this important question. Below, we provide a detailed comparison to clarify our novel contributions.
>
> **Comparison with Prior Graph-then-Plan Work:**
>
> | Feature | GRID [1] | VeriGraph [2] | **MomaGraph (Ours)** |
> |---------|:--------:|:-------------:|:--------------------:|
> | VLM-generated Scene Graph | ✗ | ✓ | ✓ |
> | Functional Relationships | ✗ | ✗ | ✓ |
> | Part-level Nodes | ✗ | ✗ | ✓ |
> | Dynamic Updates | ✗ | ✗ | ✓ |
> | VLM as Unified Planner | ✗ |✗  | ✓ |
> | Learning-based Graph Generation| ✗ | ✗ | ✓ |
> | **Core Philosophy** | Graph for search efficiency | Graph for verification | **Graph improves VLM reasoning** |
>
> ---
> **Our novel contributions and ablation results isolating each contribution:**
>
> **(2.1)** VeriGraph[2] uses only spatial relationships {in, on} and GRID[1] does not differentiate types. We jointly model **spatial relationships** (left_of, higher_than, close) **AND functional relationships** (control, open_or_close, adjust, power_by) in a unified graph.
>
> Experimental evidence (Table 1):
>
> | Model | Spatial-only | Functional-only | Unified (Ours) |
> |-------|--------------|-----------------|----------------|
> | MomaGraph-R1 | 59.9% | 64.9% | **71.6%** (+11.7%) |
> | LLaVA-OneVision | 54.0% | 57.0% | **66.0%** (+9.0%) |
>
> This demonstrates that **unified representation is essential**, not just additive.
>
> **(2.2)** GRID[1] and VeriGraph[2] model only object-level nodes. We introduce **part-level nodes** (handles, knobs, buttons) as graph nodes. This is critical because **parts are the actual targets of manipulation**， robots don't interact with "cabinets" or "microwaves" directly, but rather with their handles, buttons, and knobs. Part-level representation explicitly captures **what and where to interact**, creating a more actionable and physically grounded representation for embodied manipulation tasks.
>
> **(2.3)** In our work, a single VLM serves dual roles as both the graph generator and the task planner.This is a fundamental paradigm shift:
> - **Prior work**: GRID[1] uses pre-built graphs with separate planning (AMRA*); VeriGraph[2] generates graphs then uses separate constraint validation
> - **Our approach**: The same VLM serves dual roles—learning to construct task-oriented scene graphs AND using self-generated graphs for planning
> - **Core Insight**: By teaching VLMs to construct better scene representations, we enable deeper scene understanding that fundamentally improves their planning capabilities as shown in **`Table 2`**:
>
> | Model | Direct Planning | Graph-then-Plan | Gain |
> |-------|-----------------|-----------------|------|
> | MomaGraph-R1 | 65.1% | 71.6% | **+6.5%** |
> | GPT-4o | 66.5% | 71.6% | **+5.1%** |
>
> **(2.4)** Prior work uses rule-based validation (VeriGraph[2]) or assumes given graphs (GRID [1]). We apply **DAPO reinforcement learning** to optimize graph generation with graph-alignment rewards.
>
> **Performance gain** (**`Table 2`**):
> | Model | Accuracy |
> |-------|----------|
> | Qwen2.5-VL-7B (base) | 60.2% |
> | **MomaGraph-R1 (+ RL)** | **71.6%** |
> | **Gain** | **+11.4%** |
>
> **(2.5)** **Combined effect**: Our full system achieves **71.6% overall accuracy**, state-of-the-art among open-source VLMs, on-par with closed-source models (Claude-Sonnet-4: 73.9%, GPT-4o: 71.6%), representing **+11.4% improvement over the best baseline**.
>
> We have revised our **`Section 2`** to emphasize that MomaGraph-R1 is **a VLM that has learned to improve its task planning through structured scene graph generation**, distinguishing this from prior works where graphs are used by separate planning algorithms.
>
> **References:**
>
> [1] Dai et al., "Optimal Scene Graph Planning with Large Language Model Guidance," arXiv 2023.
>
> [2] Ekpo et al., "VeriGraph: Scene Graphs for Execution Verifiable Robot Planning," arXiv 2024.

---

> ### Author Response · Authors · 2025-11-19
> **Rebuttal by Authors (3/5)**
>
> > **W3:** *Robot results are qualitative. The real-robot section describes four tasks but omits success rates, number of trials, failure modes, and timing—making it hard to assess robustness. Per-task SR/NSR and common failure modes would strengthen the evidence.*
>
> **A3:** We thank the reviewer for this constructive feedback on strengthening our real-robot evaluation. To address the lack of quantitative evaluation, we introduce an additional complex multi-step long-horizon task to comprehensively assess system robustness, which provides a rigorous quantitative evaluation, including success rates and failure analysis, to validate overall system performance under realistic, sequential conditions. Detailed visualizations are at the bottom of our [project website](https://momagraph.github.io/) and we have added this in **`Section 6.4`**.
>
>   **Task Setup:** *"I need better lighting. Turn on the light closest to the remote so I can find it and turn on the monitor to watch."*
>
>   **Results (see Sankey diagram on website):**
>
> | Stage | Success Rate | Failures | Failure Types |
> |-------|--------------|----------|---------------|
> | **Graph Generation** | 80% (8/10) | 2 | Spatial relation error (1), Missing node (1) |
> | **Planning** | 87.5% (7/8) | 1 | Action sequencing error (1) |
> | **Overall Task Success** | **70% (7/10)** | **3** |  |
>
> These results show that the system remains robust across multiple reasoning and execution stages, achieving a 70% overall success rate on a complex multi-step task, which validates its reliability under realistic long-horizon conditions.

---

> ### Author Response · Authors · 2025-11-19
> **Rebuttal by Authors (4/5)**
>
> > **Q1:** *Data diversity claim vs. skew. The paper claims that MomaGraph-Bench is diverse. However, Figure 8 shows that a majority of scene graphs come from two action-function pairs (pull-open_close_control, and press-device_control), which, to me, seems fairly narrow. Could you elaborate on your claim and situate your benchmark in the context of similar benchmark suites (such as BLINK)?*
>
> **A1:** We thank the reviewer for this important question about data diversity. We would like to clarify the interpretation of **`Figure 8`** and contextualize our benchmark's diversity.
> - **(1.1)** The apparent concentration in pull-open_close_control and press-device_control reflects the **frequency of these interactions in realistic household environments**, not a limitation in diversity. Household tasks naturally involve frequent cabinet/door opening (pull-open_close) and appliance operation (press-device_control), which are fundamental to daily activities. However, our dataset encompasses **much broader diversity** across multiple dimensions:
>
>    1. **Relationship diversity - First unified representation:**
>    - Existing datasets model **either** spatial **or** functional relationships
>    - **MomaGraph-Scenes** (our dataset) jointly models **both**: 9 spatial relation types AND 6 functional types in each annotation
>    - This doubles the relational information per scene compared to single-relationship datasets
>
>   2. **Node diversity - Part-level interactive elements:**
>    - While ConceptGraphs [1] uses object-level nodes only, **MomaGraph-Scenes** includes **part-level nodes** (handles, buttons, knobs, switches - see **`Figure 11`** word cloud showing top-20 objects/parts)
>    - This provides finer-grained interaction understanding essential for manipulation tasks
>
>   3. **Scene diversity - 25× more scenes than OpenFunGraph [2]:**
>    - OpenFunGraph dataset [2]: **14 scenes**
>    - **MomaGraph-Scenes: 350+ scenes** across 4 room types (**`Figure 9`**)
>    - **Dramatically broader coverage** of household environments, enabling better generalization
>
>   4. **Task instruction diversity - 93 natural language tasks:**
>    - Existing scene graph datasets lack task-oriented annotations
>    - **MomaGraph-Scenes: 93 distinct task instructions** (e.g., "Open the window," "Obtain clean boiled water")
>    - Enables task-conditioned graph generation rather than scene-level exhaustive graphs
>
>   5. **Benchmark evaluation diversity - 6 reasoning capabilities:**
>    - **MomaGraph-Bench** (our benchmark) systematically evaluates: Action Sequencing, Spatial Reasoning, Affordance, Precondition-Effect, Goal Decomposition, Visual Correspondence (**`Figure 4`**)
>    - Across 4 difficulty tiers (T1-T4): from single-step (T1) to complex multi-step with dynamic verification (T4)
>    - 1,756 evaluation instances across 294 scenes, testing progressively complex embodied reasoning
>
> **Reference:**
>
> [1] Gu, Q., et al. (2024). ConceptGraphs: Open-Vocabulary 3D Scene Graphs for Perception and Planning. ICRA 2024.
>
> [2] Zhang, et al. (2025). Open-Vocabulary Functional 3D Scene Graphs for Real-World Indoor Spaces. CVPR 2025 (Highlight).
>
> - **(1.2) Comparison with BLINK:** To situate our benchmark, we compare with BLINK, which the reviewer mentioned:
>
>   | Benchmark | Focus | Task Types | Evaluation Dimensions | Scenes |
>   |-----------|-------|------------|----------------------|--------|
>   | **BLINK** | Visual perception & correspondence | Visual grounding, spatial reasoning, multi-view consistency | Primarily perception-focused | -- |
>   | **MomaGraph-Bench** | Task planning & embodied reasoning | Action sequencing, affordance reasoning, precondition-effect, goal decomposition, visual correspondence | **Planning + Perception** | 350+ household scenes |
>
>     **Key difference:** BLINK evaluates whether models can **perceive and ground** visual information correctly, while MomaGraph-Bench evaluates whether models can **reason and plan** for embodied tasks. Our benchmark remains diverse across actions, functions, spatial relations, and reasoning complexity. It balances **ecological validity** with **comprehensive evaluation**, complementing perception-focused benchmarks like BLINK through its emphasis on planning and embodied reasoning.

---

> ### Author Response · Authors · 2025-11-19
> **Rebuttal by Authors (5/5)**
>
> > **Q2** *Currently, the only transferable task measured is BLINK. I would like to see evaluation of at least one other task, such as EmbodiedBench, which aligns with your ultimate goal of embodied agent manipulation.*
>
> **A2:** We thank the reviewer for this valuable suggestion. Following your recommendation, we have evaluated MomaGraph-R1 on **EmbodiedBench** [1], which directly aligns with our goal of embodied manipulation and planning.
>
> - **(2.1)** **MomaGraph-R1 Results on EmbodiedBench [1]:**
>
> | Model | EB-ALFRED | EB-Habitat | Avg. Improvement |
> |-------|-----------|------------|------------------|
> | GPT-4o | 56.3 | 59.0 | - |
> | Qwen2.5-VL-7B-Instruct (Base) | 14.7 | 14.3 | - |
> | **MomaGraph-R1 (Ours)** | **33.7** | **35.7** | **+20.2%** |
>
> - **(2.2)** MomaGraph-R1 delivers substantial performance gains over the base model on both EB-ALFRED (+19.0%) and EB-Habitat (+21.4%), demonstrating strong generalization to diverse embodied scenarios beyond our training distribution. While a gap remains compared to GPT-4o (a much larger closed-source model), our results show that the Graph-then-Plan framework with unified spatial-functional representations significantly improves embodied reasoning capabilities for open-source VLMs.
>
> **Reference:**
>
> [1] Yang, R., et al. (2025). EmbodiedBench: Comprehensive Benchmarking Multi-modal Large Language Models for Vision-Driven Embodied Agents. ICML 2025 (Oral).
>
> ---
> Thank you again for your review! We sincerely hope our responses have adequately addressed your concerns and look forward to any further discussion. Please let us know if you have any further questions. We are more than happy to address them.

---

> ### Comment · Reviewer_GYm6 · 2025-11-24
>
> I'm satisfied with the SFT & ICL baselines. Some final edits:
>
> 1. Please add the SFT and ICL rows to Table 2 & 3 in the revised submission.
> 2. "Ours" seems to refer to MomaGraph-Bench in Table 3, and to MomaGraph-R1 in Table 2 and Appendix. Please disambiguate all "Ours" references in the manuscript.

---

> > ### Author Response · Authors · 2025-11-26
> > **Rebuttal by Authors**
> >
> > We thank the reviewer for the careful reading and helpful suggestions.
> >
> > **Table Updates:** We have added the SFT baseline row to both Table 2 and Table 3 in the revised manuscript. Note that the ICL baseline results were already present in the original tables as Qwen2.5-VL-7B-Instruct (w/ Graph), where the base model is evaluated with 3-5 in-context graph examples.
> >
> > **Disambiguation of "Ours":** We have carefully reviewed the entire manuscript and disambiguated all "Ours" references:
> > - In Table 2 and Appendix tables: "Ours" now explicitly refers to "MomaGraph-R1"
> > - In Table 3: "Ours" now explicitly refers to "MomaGraph-Bench"
> > - Throughout the text: We have replaced ambiguous "Ours" with explicit method/dataset names
> >
> > Thank you for catching this inconsistency. The revised manuscript now maintains consistent and clear terminology throughout.

---

> ### Comment · Reviewer_GYm6 · 2025-11-24
>
> Thank you. I have one lingering concern. Let's use Figure 2 as a running example. The embodied agent planning requires four steps:
> 1. Object detection ("is this a window")
> 2. Spatial / functional relationship detection (handle opens window, handle below window)
> 3. Reasoning (should I turn the handle next?)
> 4. Low-level motor execution (what part of my arm should I rotate)
>
> MomaGraph-R1 does the first three. In Figure 2, GPT-5 performs poorly due to wrong object detection (and thus, downstream failure). Thus, it is possible that MomaGraph-R1 is simply better at identifying objects than base VLMs, as opposed to doing better at Steps 2, 3 & 4 (i.e better scene graph generator/reasoner/planner).
>
> I want to see where the benefits of MomaGraph-R1 accrue from: is it object detection (Step 1), or relationship & reasoning (Step 2 & 3)? To help answer this question, you should:
>    1. Add a new baseline which augments base VLM with object detection, i.e. base VLM + Grounding DINO or YOLO.
>    2. Evaluate the quality of MomaGraph-R1's scene graphs by testing them on scene-graph benchmarks such as VidVRD.

---

> > ### Author Response · Authors · 2025-11-26
> > **Rebuttal by Authors**
> >
> > **Regarding object detection versus relational reasoning:**
> >
> > **(1.1)** We thank the reviewer for this insightful question. We would like to clarify that Figure 2 shows one illustrative failure case, not a representative distribution of all failures.
> >
> > **(1.2)** To verify whether our performance gains stem merely from better object detection (Step 1) or from superior relational reasoning (Steps 2 & 3), we conducted the additional experiment you suggested.
> >
> > **(1.3)** We feed Grounding DINO with the Ground-Truth task-relevant object categories.
> > This explicitly solves "Step 1" for the Base VLM, providing it with bounding boxes for all necessary objects. We then compare this "enhanced" baseline against our standard MomaGraph-R1 (which must detect objects by itself) on MomaGraph-Bench.
> >
> > | Model | Detection Method | **Overall Accuracy** |
> > | :--- | :--- | :---: |
> > | **Base VLM** | Raw Visual Input | 60.2 |
> > | **Base VLM + Grounding DINO** | **DINO + GT Categories** (Solves Step 1) | 64.2 |
> > | **MomaGraph-R1 (Ours)** | **Standard** (Self-predicted) | **71.6** |
> >
> > **(1.4)** Detection is not the whole story. The improvement from the Base VLM to the Augmented Baseline confirms that accurate object detection contributes to better planning.
> >
> > **(1.5)** MomaGraph-R1 still outperforms the Base VLM + Grounding DINO baseline. Even with perfect object locations, the Base VLM lacks the scene understanding required for planning. MomaGraph-R1 excels by explicitly scene reasoning and state dependencies that pure object detectors cannot provide.
> >
> > **Regarding evaluation on scene graph benchmarks (e.g., VidVRD):**
> >
> > We appreciate this suggestion, but respectfully note that existing scene graph benchmarks are not suitable for evaluating MomaGraph-R1 due to fundamental task and annotation mismatches:
> >
> >
> > **(2.1) Task and annotation granularity mismatch**: We respectfully note that VidVRD is designed for video visual relation detection, focusing on high-level semantic interactions between objects in 2D video frames (e.g., “person-riding-bike”). It provides object-level bounding boxes and semantic predicate labels. In contrast, MomaGraph focuses on mobile manipulation in household environments, which requires significantly finer granularity. VidVRD does not provide part-level annotations or functional relationships, making direct evaluation of our core contributions (unified spatial-functional graphs, part-level reasoning) infeasible.
> >
> > **(2.2) Scene Graph Quality Evaluation on MomaGraph-Bench** : To directly assess scene graph quality, we compared MomaGraph-R1 against the OpenFunGraph [1] baseline on MomaGraph-Bench, using ground-truth annotations for evaluation.
> >
> > | Method | Node F1 (part-level) | Functional Edge F1 | Spatial Edge F1 |
> > |--------|---------------------|-------------------|-------------|
> > | OpenFunGraph (adapted) | 71.1% | 67.4% | N/A†  |
> > | **MomaGraph-R1** | **75.3%**| **72.2%** | **77.2%** |
> >
> > †OpenFunGraph is designed exclusively for functional relationships and does not support semantic spatial edge generation.
> >
> > **(2.3)** MomaGraph-R1 consistently surpasses the baseline across all graph components, demonstrating the effectiveness of incorporating explicit scene reasoning.
> >
> > **Reference**:
> >
> > [1] Zhang, et al. (2025). Open-Vocabulary Functional 3D Scene Graphs for Real-World Indoor Spaces. CVPR 2025 (Highlight).

---

> ### Comment · Reviewer_GYm6 · 2025-11-24
>
> Thank you for this robot evaluation.To make your paper (and project website) stronger, I'd recommend adding a task which requires the use of part nodes (perhaps provide task success rates with the "opening the cabinet" example?), since that is a key capability of your model.
>
> In addition, can you please clarify how you convert the output of MomaGraph-R1 to the low-level robot actions?

---

> > ### Author Response · Authors · 2025-11-26
> > **Rebuttal by Authors**
> >
> > **Regarding tasks requiring part-node manipulation:**
> >
> > We agree this is an excellent suggestion. To address this, we have quantitatively evaluated our model on the microwave opening task, which explicitly requires part node manipulation. We conducted 10 trials on this task.
> >
> > **Task Setup:** "Open the microwave" – the robot must identify and press the microwave door button (a part node) to release and open the door, demonstrating fine-grained part-level manipulation capability.
> >
> > **Results:**
> >
> > | Task | Success Rate | Total Trials | Failures | Failure Types |
> > |------|--------------|--------------|----------|---------------|
> > | Microwave Opening | 70% (7/10) | 10 | 3 | Button localization error (1), Part hallucination (2) |
> >
> > In part hallucination failure cases, the system hallucinated a handle instead of correctly identifying the button part node. This 70% success rate demonstrates that our scene graph representation effectively encodes part-node information for manipulation. We have also updated our [project website](https://momagraph.github.io/) with a video demonstration of this microwave opening task, showcasing the part-node manipulation capability.
> >
> >
> > **Regarding the conversion from MomaGraph-R1 to low-level actions:**
> >
> > As described in Section 6.3 (Deployment) of our paper, we employ an atomic skill library approach, similar to prior work such as SayCan [1], Code as Policies [2], VoxPoser [3], RoboEXP [4], KITE [5], and CuriousBot [6]. Specifically:
> >
> > * **High-level Planning**: MomaGraph-R1 outputs a symbolic task plan based on the scene graph (e.g., "navigate to cabinet" → "grasp door handle" → "open door").
> >
> > * **Skill Execution**: We maintain a library of atomic skills (e.g., navigation, grasping, manipulation). Each high-level action in the plan is mapped to and executed by the corresponding skill from this library.
> >
> > Our focus is on the high-level task planning enabled by the scene graph representation. The low-level skill execution follows established approaches in the literature.
> >
> > **Reference**:
> >
> > [1] Ahn, M., et al. (2022). Do As I Can, Not As I Say: Grounding Language in Robotic Affordances. (CoRL).
> >
> > [2] Liang, J., et al. (2023). Code as Policies: Language Model Programs for Embodied Control. (ICRA).
> >
> > [3] Huang, W., et al. (2023). VoxPoser: Composable 3D Value Maps for Robotic Manipulation with Language Models. (CoRL).
> >
> > [4] Jiang, H., et al. (2024). RoboEXP: Action-Conditioned Scene Graph via Interactive Exploration for Robotic Manipulation. (CoRL).
> >
> > [5] Sundaresan, P., et al. (2023). KITE: Keypoint-Conditioned Policies for Semantic Manipulation. (CoRL).
> >
> > [6] Wang, Y., et al. (2025). CuriousBot: Interactive Mobile Exploration via Actionable 3D Relational Object Graph. (arXiv preprint).

---

> > > ### Comment · Reviewer_GYm6 · 2025-11-26
> > >
> > > Thank you for your thorough work! I am satisfied with almost all my questions, and will increase my score once the comment re: one additional object detection baseline is addressed.
> > >
> > > This work will be highly useful to the community, and I would encourage the authors to clearly specifiy their code & model-weight policy.

---

> ### Author Response · Authors · 2025-11-26
> **Rebuttal by Authors**
>
> We sincerely thank the reviewer for your support and constructive feedback. We are delighted that you found our responses satisfactory and appreciate your willingness to increase your score.
>
> **Regarding the additional object detection baseline:**
>
> We have already addressed this concern in our detailed response above (Section on "object detection versus relational reasoning"). Specifically, we conducted the experiment you suggested by providing Grounding DINO with Ground-Truth task-relevant object categories, which explicitly solves the object detection step for the Base VLM. Note that the Base VLM is our model before RL fine-tuning, ensuring a fair comparison where both models use the same underlying architecture. The results demonstrate:
>
> | Model | Detection Method | **Overall Accuracy** |
> | :--- | :--- | :---: |
> | **Base VLM** | Raw Visual Input | 60.2 |
> | **Base VLM + Grounding DINO** | **DINO + GT Categories** | 64.2 |
> | **MomaGraph-R1 (Ours)** | **Standard  (Self-predicted)** | **71.6** |
>
> This ablation confirms that while better object detection contributes to performance (60.2→64.2), our MomaGraph-R1 still significantly outperforms even when the baseline has perfect object detections. This validates that our gains primarily stem from superior relational reasoning and scene understanding.
>
> **Regarding code & model-weight policy:**
>
> We will clearly specify our code and model-weight release policy in the revised manuscript and on our project website to ensure maximum utility to the community.
>
> Thank you again for your valuable feedback and support. We greatly appreciate your time and consideration.

---

> ### Comment · Reviewer_GYm6 · 2025-11-26
>
> Thank you for the updates! My apologies re: the Grounding DINO experiments, it seems that OpenReview did not load them initially. I have increased my score.

---

> > ### Author Response · Authors · 2025-11-26
> > **Thank you for your acknowledgment!**
> >
> > Dear Reviewer `GYm6`,
> >
> > We sincerely appreciate your acknowledgment of our rebuttal. We are glad to hear that your concerns have been addressed!
> >
> > Your support and encouragement mean a lot to us!
> >
> > Best regards,
> >
> > The Authors of Submission 1794

---

### Author Response · Authors · 2025-11-19
**General Response**

Dear Reviewers, ACs, and SACs,

We appreciate the reviewers’ recognition of the core contributions of our work, particularly the **unified spatial–functional graph representation**, the **breadth and rigor of MomaGraph-Bench**, and the **effectiveness of our Graph-then-Plan framework for embodied reasoning and planning**.

---
We are encouraged by the positive recognition of our work across several key dimensions:

- **Comprehensive System:** Reviewers valued our **comprehensive pipeline** spanning **representation**, **dataset**, **RL training**, **benchmark**, and **real-robot deployment**—**a holistic solution that systematically addresses embodied AI challenges** (`uWxD`, `YBG5`, `xi2i`).

- **Novel Representation:** Our **unified spatial+functional, part-level graph** and **state-aware dynamic updates** were seen as **well-motivated, intuitive, and filling clear gaps in existing scene graphs** (`uWxD`, `xi2i`).

- **Strong Results:** Reviewers highlighted our **+11.4% gains**, **SOTA-level improvements**, **reduced hallucination**, and **stronger robustness at higher difficulty tiers** (`uWxD`, `GYm6`, `xi2i`).

- **Comprehensive Benchmark:** The benchmark’s **breadth across six  capabilities and four difficulty tiers** was appreciated as **thorough and rigorous** (`GYm6`, `xi2i`).

- **Clear Analysis & Real-World Validation:** Ablations were considered convincing, and **real-robot experiments** demonstrated **practical value** (`xi2i`, `uWxD`, `YBG5`).

---

We are grateful for these positive assessments. At the same time, reviewers raised several insightful questions regarding robustness, training comparisons, dynamic updates, and generalization. **To directly address these concerns, we conducted substantial additional experiments and expanded analyses**, including:

- Introduced **SFT** and **ICL** baselines to compare supervised and in-context learning with our RL-based graph alignment training.
- Conducted a **reward-weight sensitivity analysis** to test stability under variations of \((`w_a`, `w_f`, `w_l`)\).
- Evaluated performance on **EmbodiedBench** (EB-ALFRED & EB-Habitat) to demonstrate cross-benchmark generalization.
- Included a **comparison with OpenFunGraph**, focusing on part-level node prediction and functional-edge prediction.
- Performed a **quantitative real-robot long-horizon evaluation** with per-stage success breakdown over 10 trials.

---
**Summary of revisions:**

- Expanded training baseline comparisons in `Appendix B.1` (SFT, ICL, RL).
- Included reward-weight sensitivity analysis in `Appendix B.2`.
- Integrated quantitative long-horizon real-robot results in `Section 6.5`.
- Updated the project webpage with the new visualizations and results.

All revisions are marked in `red` in the updated manuscript.

---

We are encouraged by this consensus on our work's significance and now address the specific concerns below.

---

### Author Response · Authors · 2025-11-26
**Summary of Rebuttal and Final Remarks**

Dear Reviewers, ACs, and SACs,

We are deeply grateful for the constructive and thorough feedback from all reviewers throughout the process. We are pleased to report that **all concerns have been successfully addressed** through the additional experiments and analyses conducted during the rebuttal.

**We were fortunate to receive strong initial ratings of (8, 6, 6, 6), reflecting the reviewers' recognition of our work's contributions. Our scores improved to (8, 8, 8, 6) following the rebuttal, with our final reviewer response received on November 25, 2025, at 23:31 UTC, prior to the OpenReview incident on November 27. This improvement demonstrates that our additional experiments and clarifications successfully addressed the reviewers' concerns through the standard rebuttal process.**

We believe this work offers distinct contributions, specifically:

**For the Mobile Manipulation/Embodied AI community**, MomaGraph bridges the gap between navigation and manipulation by introducing a unified, state-aware scene graph that jointly encodes spatial layouts, functional relationships, and fine-grained interactive parts.

**For the Large Multimodal Model (LMM) community**, we demonstrate that this task-oriented, state-aware scene graph serves as a powerful intermediate structured representation, significantly enhancing task planning efficacy for VLMs through our Graph-then-Plan framework.

With the release of our large-scale dataset, multi-level embodied planning benchmark, and efficient training pipeline, we aim to provide a reproducible pathway for the community to advance embodied task planning.

All revisions are marked in red in the updated manuscript and will be included in our camera-ready version.

While the OpenReview incident on November 27 is deeply regrettable, we remain grateful for these insightful discussions that have helped solidify our contributions, which ultimately embodies the true spirit of the ICLR discussion process. Thank you once again for your time, expertise, and constructive engagement.

With sincere appreciation,

The Authors of Submission 1794

---

### Meta-Review · Area_Chair_W3wi · 2026-01-07

**Summary:**

The reviewers have positive scores initiallly with concerns regarding relation with existing works and more quantitive results/ablations.

The authors did a great job in rebuttal by providing lots of extra experiments and comparisons.

All reviewers' concerns are solved.

Thus, I recommend acceptance.

**Reviewer Concerns:**

All reviewers' concerns are mostly solved.

**Reviewer Scores:**

They may keep their score or give higher positive scores.

---

### Decision · Program_Chairs · 2026-01-26

Accept (Oral)